# MultiModN—Multimodal, Multi-Task, Interpretable Modular Networks

**Vinitra Swamy***
EPFL
vinitra.swamy@epfl.ch

**Malika Satayeva***
EPFL
malika.satayeva@epfl.ch

**Jibril Frej**
EPFL
jibril.frej@epfl.ch

**Thierry Bossy**
EPFL
thierry.bossy@epfl.ch

**Thijs Vogels**
EPFL
thijs.vogels@epfl.ch

**Martin Jaggi**
EPFL
martin.jaggi@epfl.ch

**Tanja Käser***
EPFL
tanja.kaser@epfl.ch

**Mary-Anne Hartley***
Yale, EPFL
mary-anne.hartley@yale.edu

## Abstract

Predicting multiple real-world tasks in a single model often requires a particularly diverse feature space. Multimodal (MM) models aim to extract the synergistic predictive potential of multiple data types to create a shared feature space with aligned semantic meaning across inputs of drastically varying sizes (i.e. images, text, sound). Most current MM architectures fuse these representations in parallel, which not only limits their interpretability but also creates a dependency on modality availability. We present `MultiModN`, a multimodal, modular network that fuses latent representations in a sequence of any number, combination, or type of modality while providing granular real-time predictive feedback on any number or combination of predictive tasks. `MultiModN`'s composable pipeline is interpretable-by-design, as well as innately multi-task and robust to the fundamental issue of biased missingness. We perform four experiments on several benchmark MM datasets across 10 real-world tasks (predicting medical diagnoses, academic performance, and weather), and show that `MultiModN`'s sequential MM fusion does not compromise performance compared with a baseline of parallel fusion. By simulating the challenging bias of missing not-at-random (MNAR), this work shows that, contrary to `MultiModN`, parallel fusion baselines erroneously learn MNAR and suffer catastrophic failure when faced with different patterns of MNAR at inference. To the best of our knowledge, this is the first inherently MNAR-resistant approach to MM modeling. In conclusion, `MultiModN` provides granular insights, robustness, and flexibility without compromising performance.

## 1 Introduction

The world is richly multimodal and intelligent decision-making requires an integrated understanding of diverse environmental signals, known as embodied intelligence [1]. Until recently, advances in deep learning have been mostly compartmentalized by data modality, creating disembodied domains such as computer vision for images, natural language processing for text, and so on. Multimodal (MM) learning has emerged from the need to address real-world tasks that cannot be robustly represented by a single signal type as well as the growing availability and diversity of digitized signals [2, 3, 4]. Some examples are diagnosis from a combination of medical tests and imagery [5, 6, 7], estimating sentiment from facial expression, text, and sound [8, 9, 10, 11], and identifying human activities from a combination of sensors [12].

---

* denotes equal contribution

37th Conference on Neural Information Processing Systems (NeurIPS 2023).

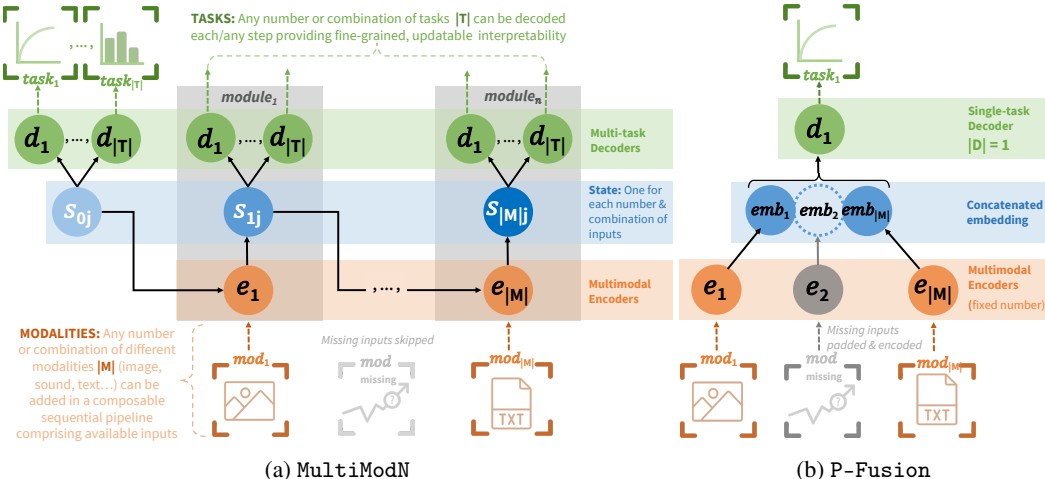

Figure 1: Comparison of modular `MultiModN` **(a)** vs. monolithic `P-Fusion` **(b)**. `MultiModN` inputs any number/combination of modalities (**mod**) into a sequence of $mod$-specific encoders (**e**). It can skip over missing modalities. A state (**s**) is passed to the subsequent encoder and updated. Each state can be fed into any number/combination of decoders (**d**) to predict multiple tasks. *Modules* are identified as grey blocks comprising an encoder, a state, and a set of decoders. `P-Fusion` is a monolithic model. It inputs a *fixed* number/combination of modalities (**mod**) into $mod$-specific encoders (**e**). Missing modalities are padded and encoded. Embeddings (**emb**) are concatenated and provided to a *single* decoder in parallel (**d**) to predict a *single* task.

The richer representations from synergistic data types also have the potential to increase the task space, where a single set of representations can generalize to several tasks. Multi-task (MT) learning has not only been shown to benefit the performance of individual tasks but also has the potential to greatly reduce computational cost through shared feature extraction [13].

In short, multimodal and multi-task learning hold significant potential for human-centric machine learning and can be summarized respectively as creating a shared feature space from various data types and deriving their relative semantic meaning across several tasks.

**Limitations of current multimodal models.** Current MM models propose a parallel integration of modalities, where representations are fused and processed simultaneously [2, 3, 4]. Parallel fusion (hereafter `P-Fusion`) creates several fundamental limitations that we address in this work.

The most important issue we seek to resolve in current MM architectures is their *dependence on modality availability* where all modalities for all data points are required inputs during both training and inference. Modality-specific missingness is a common real-world problem and can fundamentally bias the model when the missingness of a modality is predictive of the label (known as missing not-at-random, MNAR). The common solution of restricting learning to data points with a complete set of modalities creates models that perform inequitably in populations with fewer available resources (i.e. when the pattern of MNAR is different between train and test sets). In complex real-world datasets, there is often no intersection of complete availability, thus necessitating the exclusion of modalities or significantly limiting the train set. On the other hand, imputation explicitly featurizes missingness, thus risking to create a trivial model that uses the *presence* of features rather than their value for the prediction [14, 15]. The MNAR issue is particularly common in medicine, where modality acquisition is dependent on the decision of the healthcare worker (i.e. the decision that the model is usually attempting to emulate). For example, a patient with a less severe form of a disease may have less intensive monitoring and advanced imagery unavailable. If the goal is to predict prognosis, the model could use the missingness of a test rather than its value. This is a fundamental flaw and can lead to catastrophic failure in situations where the modality is not available for independent reasons (for instance resource limitations). Here, the featurized missingness would inappropriately categorize the patient in a lower severity class. For equitable real-world predictions, it is critical to adapt predictions to available resources, and thus allow composability of inputs at inference.

Another key issue of current techniques that this work addresses is *model complexity*. Parallel fusion of various input types into a single vector make many post-hoc interpretability techniques difficult

or impossible [16]. Depending on where the fusion occurs, it may be impossible to decompose modality-specific predictive importance.

In this work, we leverage network modularization, compartmentalizing each modality and task into independent encoder and decoder modules that are inherently robust to the bias of MNAR and can be assembled in any combination or number at inference while providing continuous modality-specific predictive feedback.

**Contributions.** We propose `MultiModN`, a multimodal extension of the work of Trottet et al. [17], which uses a flexible sequence of model and task-agnostic encoders to produce an evolving latent representation that can be queried by any number or combination of multi-task, model-agnostic decoder modules after each input (showcased in Figure 1). Specifically, we demonstrate that our modular approach of sequential MM fusion:

[1] **matches parallel MM fusion** (`P-Fusion`) for a range of real-world tasks across several benchmark datasets, while contributing distinct advantages, such as being:

[2] **composable at inference**, allowing selection of any number or combination of available inputs,

[3] **is robust to the bias of missing not-at-random (MNAR) modalities**,

[4] **is inherently interpretable**, providing granular modality-specific predictive feedback, and

[5] **is easily extended to any number or combination of tasks**.

We provide an **application-agnostic open-source framework** for the implementation of `MultiModN`: https://github.com/epfl-iglobalhealth/MultiModN. Our experimental setup purposely limits our model performance to fairly compare the multimodal fusion step. At equivalent performance, our model architecture is by far superior to the baseline by virtue of being inherently modular, interpretable, composable, robust to systematic missingness, and multi-task.

## 2   Background

Approaches to MM learning can be categorized by the depth of the model at which the shared feature space is created [2]. Late fusion (decision fusion) processes inputs in separate modality-specific sub-networks, only combining the outputs at the decision-level, using a separate model or aggregation technique to make a final prediction. While simple, late fusion fails to capture relationships between modalities and is thus not *truly* multimodal. Early fusion (feature fusion), combines modalities at the input level, allowing the model to learn a joint representation. Concatenating feature vectors is a popular and simple approach [18, 19], but the scale of deployment is particularly limited by the curse of dimensionality. Finally, intermediate fusion (model fusion) seeks to fine-tune several feature extraction networks from the parameters of a downstream classifier.

**Parallel Multimodal Fusion** (`P-Fusion`).    Recently, Soenksen et al. [20] proposed a fusion architecture which demonstrated the utility of multiple modalities in the popular MM medical benchmark dataset, MIMIC [21, 22]. Their framework (HAIM or Holistic Artificial Intelligence in Medicine) generates single-modality embeddings, which are concatenated into a single one-dimensional multimodal fusion embedding. The fused embedding is then fed to a single-task classifier. This work robustly demonstrated the value of multimodality across several tasks and a rich combination of heterogeneous sources. HAIM consistently achieved an average improvement of 6-33% AUROC (area under the receiver operating characteristic curve) across all tasks in comparison to single-modality models. We use this approach as a `P-Fusion` baseline against our sequential fusion approach of `MultiModN` and extend it to several new benchmark datasets and multiple tasks.

Soenksen et al. [20] perform over 14,324 experiments on 12 binary classification tasks using every number and combination of modalities. This extreme number of experiments, was necessary because the model is not composable nor capable of multi-task (MT) predictions. Rather, a different model is needed for each task and every combination of inputs for each task. In contrast, `MultiModN` is an extendable network, to which any number of encoders and decoders can be added. Thus, most of the 14,324 experiments could technically be achieved within one `MultiModN` model.

Several other recent architectures utilize parallel fusion with transformers. UNiT (Unified Transformer) [23] is a promising multimodal, multi-task transformer architecture; however, it remains monolithic, trained on the union of all inputs (padded when missing) fed in parallel. This not only exposes the model to patterns of systematic missingness during training but also reduces model

interpretability and portability[1]. [24]'s recent work has found similar results on the erratic behavior of transformers to missing modalities, although it is only tested on visual/text inputs. LLMs have also recently been used to encode visual and text modalities [25], but it is not clear how tabular and time-series would be handled or how this would affect the context window at inference. Combining predictive tasks with LLMs will also greatly impact interpretability, introducing hallucinations and complex predictive contamination where learned textual bias can influence outcomes.

**Modular Sequential Multimodal Fusion.**   A *module* of a modular model is defined as a self-contained computational unit that can be isolated, removed, added, substituted, or ported. It is also desirable for modules to be order invariant and idempotent, where multiple additions of the same module have no additive effect. We design `MultiModN` to encode individual inputs, whereby module-exclusion can function as input *skippablity*, allowing missing inputs to be skipped without influencing predictions. Thus, modular models can have various input granularities, training strategies, and aggregation functions. Some popular configurations range from hierarchies with shared layers to ensemble predictions and teacher-trainer transfer learning approaches [26, 27].

We expand on the sequential modular network architecture proposed by Trottet et al.[17] called MoDN (Modular Decision Networks) as a means of sequential MM fusion. MoDN trains a series of feature-specific encoder modules that produce a latent representation of a certain size (the *state*). Modules can be strung together in a mix-and-match sequence by feeding the state of one encoder as an input into the next. Therefore, the state has an additive evolution with each selected encoder. A series of decoders can query the state at any point for multiple tasks from various combinations of inputs, giving MoDN the property of combinatorial generalization.

Thus, we extend MoDN to learn multiple tasks from multimodal inputs. By aligning feature extraction pipelines between `MultiModN` and the `P-Fusion` baseline (inspired by HAIM) we can achieve a better understanding of the impact of monolithic-parallel fusion vs. sequential-modular MM fusion. Figure 1 provides a comparison between `P-Fusion` and `MultiModN`, also formalized below.

## 3   Problem formulation

**Context.** We propose a multi-task supervised learning framework able to handle any number or combination of inputs of varying dimension, irrespective of underlying bias in the availability of these inputs during training. We modularize the framework such that each input and task is handled by distinct encoder and decoder *modules*. The inputs represent various data modalities (i.e. image, sound, text, time-series, tabular, etc.). We assume that these inputs have synergistic predictive potential for a given target and that creating a multimodal shared feature space will thus improve model performance. The tasks represent semantically related observations. We hypothesize that jointly training on semantically related tasks will inform the predictions of each individual task.

**Notation.** Formally, given a set of modalities (features) $\mathcal{M} = \{mod_1, \ldots, mod_{|\mathcal{M}|}\}$ and a set of tasks (targets) $\mathcal{T} = \{task_1, \ldots, task_{|\mathcal{T}|}\}$, let $\mathcal{X} = \{(x_1, y_1), (x_2, y_2), \ldots, (x_N, y_N)\}$ represent a multimodal, multi-task dataset with $N$ data points $(x_1, \ldots, x_N)$.

Each point $x$ has $|\mathcal{M}|$ modalities (inputs): $x = \{x_{mod_1}, \ldots, x_{mod_{|\mathcal{M}|}}\}$ and is associated with a set of $|\mathcal{T}|$ targets (tasks): $y = \{y_{task_1}, \ldots, y_{task_{|\mathcal{T}|}}\}$. Modalities comprise various sources (e.g. images from x-rays, CT), for simplicity, we consider sources and modalities as equal $mod$ elements in $\mathcal{M}$.

**Multimodal, multi-task, modular formulation.** We decompose each data point $x$ into $|\mathcal{M}|$ sequential encoder *modules* specific to its constituent modalities and each target $y$ into $|\mathcal{T}|$ decoder *modules* specific to its constituent tasks such that any combination or number of modalities can be used to predict any combination or number of tasks. Our objective is to learn a set of function *modules*, $\mathcal{F}$. Each function *module* within this set, represented as $f_j^i \in \mathcal{F}$ maps combinations of modalities $\mathcal{M}_j$ to combinations of tasks $\mathcal{T}_i$, i.e. $f_j^i : \mathcal{M}_j \to \mathcal{T}_i$. It is important to note that $\mathcal{M}_j$ is an element of the powerset of all modalities and $\mathcal{T}_i$ is an element of the powerset of all tasks.

**Extension to time-series.** In the above formulation, the $\mathcal{M}$ encoder *modules* are handled in sequence, thus naturally aligning inputs with time-series. While the formulation does not change for time-series data, it may be optimized such that $f_j^i$ represents a single time step. This is relevant in the real-world

---

[1]The equivalent transformer architecture has 427,421 trainable parameters for the EDU dataset (Sec. 5) while `MultiModN` achieves better performance with 12,159 parameters.

setting of a data stream, where inference takes place at the same time as data is being received (i.e. predicting student performance at each week of a course as the course is being conducted). The continuous prediction tasks (shown for EDU and Weather in Sec. 6) demonstrate how `MultiModN` can be used for incremental time-series prediction.

# 4    `MultiModN`: Multimodal, Multi-task, Modular Networks (Our model)

Building on [17] (summarized and color-coded in Figure 1a), the `MultiModN` architecture consists of three modular elements: a set of **State** vectors $\mathcal{S} = \{s_0, \ldots, s_{|\mathcal{M}|}\}$, a set of modality-specific **Encoders** $\mathcal{E} = \{e_1, \ldots, e_{|\mathcal{M}|}\}$, and a set of task-specific **Decoders** $\mathcal{D} = \{d_1, \ldots, d_{|\mathcal{T}|}\}$. State $s_0$ is randomly initialized and then updated sequentially by $e_i$ to $s_i$. Each $s_i$ can be decoded by one, any combination, or all elements of $\mathcal{D}$ to make a set of predictions. All encoder and decoder parameters are subject to training.

**States ($\mathcal{S}$).** Akin to hidden state representations in Recurrent Neural Networks (RNNs), the state of `MultiModN` is a vector that encodes information about the previous inputs. As opposed to RNNs, state updates are made by any number or combination of modular, modality-specific encoders and each state can be decoded by modular, task-specific decoders. Thus the maximum number of states by any permutation of $n$ encoders is $n!$. For simplicity, we limit the combinatorial number of states to a single order (whereby $e_i$ should be deployed before $e_{i+1}$) in which any number or combination of encoders may be deployed (i.e. one or several encoders can be skipped at any point) as long as the order is respected. Thus, the number of possible states for a given sample is equal to $2^{|\mathcal{M}|}$. Order invariance could be achieved by training every permutation of encoders $|\mathcal{M}|!$, i.e. allowing encoders to be used in any order at inference, as opposed to this simplified implementation of `MultiModN` in which order is fixed. At each step $i$, the encoder $e_i$ processes the previous state, $s_{i-1}$, as an input and outputs an updated state $s_i$ of the same size. When dealing with time-series, we denote $s_{t(0)}$ as the state representing time $t$ before any modalities have been encoded, and as $s_{t(0,1,4,5)}$ as the state at time $t$ after being updated by encoders $e_1$, $e_4$ and $e_5$, in that order.

**Encoders ($\mathcal{E}$).** Encoders are modularized to represent a single modality, i.e. $|\mathcal{E}| = |\mathcal{M}|$. An encoder $e_i$ takes as input the combination of a single modality (of any dimension) and the previous state $s_{i-1}$. Encoder $e_i$ then outputs a $s_i$, updated with the new modality. For simplicity, we fix the state size between encoders. Due to modularization, `MultiModN` is model-agnostic, whereby encoders can be of any type of architecture (i.e. Dense layers, LSTM, CNN). For experimental simplicity, we use a single encoder design with a simple dense layer architecture. The input vectors in our experiments are 1D. When a modality is missing, the encoder is skipped and not trained (depicted in Figure 1).

**Decoders ($\mathcal{D}$).** Decoders take any state $s_i$ as input and output a prediction. Each decoder is assigned to a single task, that is $|\mathcal{D}| = |\mathcal{T}|$, i.e. `MultiModN` is not multiclass, but multi-task (although a single task may be multiclass). Decoders are also model-agnostic. Our implementation has regression, binary, and multiclass decoders across static targets or changing time-series targets. Decoder parameters are shared across the different modalities. The decoder predictions are combined across modalities/modules by averaging the loss. Interestingly, a weighted loss scheme could force the model to emphasize certain tasks over others.

As shown in [17], `MultiModN` can be completely order-invariant and idempotent if randomized during training. For interpretability, sequential inference (in any order) is superior to parallel input due to its decomposability, allowing the user to visualize the effect of each input and aligning with Bayesian reasoning.

**Quantification of Modularity.** The modularity of a network can be quantified, whereby neurons are represented by nodes (vertices) and connections between neurons as edges. There are thus comparatively dense connections (edges) within a *module* and sparse connections between them. Partitioning modules is an NP-complete problem [28]. We present modules that are defined *a priori*, whereby a module comprises one encoder $e_i$ connected to one state $s_i$, which is in turn connected to a set of $|\mathcal{T}|$ tasks (a *module* is depicted as a grey box in Figure 1a). Following a formalization of modularity quantitation proposed by Newman et al. [29], we compute the modularity score for `MultiModN` and show that it tends to a perfect modularity score of 1 with each added modality and each added task. When viewed at the network granularity of these core elements, P-Fusion is seen as a monolithic model with a score of 0. The formula is elaborated in Appendix Sec. B.

## 4.1 `P-Fusion`: Parallel Multimodal Fusion (Baseline)

We compare our results to a recent multimodal architecture inspired by HAIM (Holistic AI in Medicine) [20]. As depicted in Figure 1b, HAIM also comprises three main elements, namely, a fixed set of modality-specific encoders $\mathcal{E} = \{e_1, \ldots, e_{|\mathcal{M}|}\}$ which create a fixed set of embeddings $\mathcal{B} = \{emb_1, \ldots, emb_{|\mathcal{M}|}\}$, that is concatenated and fed into a single-task decoder ($d_1$). HAIM achieved state-of-the-art results on the popular and challenging benchmark MIMIC dataset, showing consistently that multimodal predictions were between 6% and 33% better than single modalities.

**Encoders ($\mathcal{E}$).** Contrary to the flexible and composable nature of `MultiModN`, the sequence of encoders in `P-Fusion` is fixed and represents a unique combination of modalities. It is thus unable to skip over modalities that are missing, instead padding with neutral values and explicitly embedding the non-missing modalities. The encoders are modality-specific pre-trained neural networks.

**Embeddings ($\mathcal{B}$).** Multimodal embeddings are fused in parallel by concatenation.

**Decoders ($\mathcal{D}$).** Concatenated embeddings are passed to a single-task decoder.

**Architecture alignment.** We align feature extraction between `MultiModN` and `P-Fusion` to best isolate the effect of sequential (`MultiModN`) vs. parallel (`P-Fusion`) fusion. As depicted in Appendix Figure 8, we let `MultiModN` take as input the embeddings created by the `P-Fusion` pre-trained encoders. Thus both models have identical feature extraction pipelines. No element of the `MultiModN` pipeline proposed in Figure 1a is changed. The remaining encoders and decoders in both models are simple dense layer networks (two fully connected ReLU layers and one layer for prediction). Importantly, `MultiModN` encoders and decoders are model-agnostic and can be of any architecture.

## 5 Datasets

We compare `MultiModN` and `P-Fusion` on three popular multimodal benchmark datasets across 10 real-world tasks spanning three distinct domains (healthcare, education, meteorology). The healthcare dataset (MIMIC) is particularly challenging in terms of multimodal complexity, incorporating inputs of vastly varying dimensionality. Education (EDU) and Weather2k have a particular focus on time-series across modalities. Appendix Sec. C details features, preprocessing, and tasks ($task_{1-10}$).

**MIMIC.** MIMIC [30] is a set of deidentified electronic medical records comprising over $40,000$ critical care patients at a large tertiary care hospital in Boston. The feature extraction pipeline replicated according to our baseline of `P-Fusion` [20], making use of patient-level feature embeddings extracted from pre-trained models as depicted in Appendix Figure 8. We select the subcohort of 921 patients who have valid labels for both diagnoses and all four modalities present. We use all four modalities as inputs: chest x-rays (image), chart events (time-series), demographic information (tabular), and echocardiogram notes (text). For simplicity, we focus on two diagnostic binary classification tasks: cardiomegaly ($task_1$) and enlarged cardiomediastinum ($task_2$). These tasks were selected for their semantic relationship and also because they were reported to benefit from multimodality [20]. Thus, we have four modality-specific encoders and two binary classification diagnostic decoders.

**Education (EDU).** This educational time-series dataset comprises $5,611$ students with over 1 million interactions in a 10-week Massively Open Online Course (MOOC), provided to a globally diverse population. It is benchmarked in several recent works [31, 32, 33]. Our modeling setting is replicated from related literature, with $45$ handcrafted time-series features regarding problem and video modalities extracted for all students at each weekly time-step [34]. We use two modality-specific encoders (problem and video) and three popular decoder targets: binary classifiers ($task_{3-4}$) of pass/fail and drop-out, and a continuous target of next week's performance ($task_5$) [35].

**Weather2k.** *Weather2k* is a 2023 benchmark dataset that combines tabular and time-series modalities for weather forecasting [36]. The data is extracted from $1,866$ ground weather stations covering 6 million $km^2$, with 20 features representing hourly interactions with meteorological measurements and three static features representing the geographical location of the station. We create five encoders from different source modalities: geographic (static), air, wind, land, and rain and align with the benchmark prediction targets [36] on five continuous regression targets: short (24 hr), medium (72 hr), long term (720 hr) temperature forecasting, relative humidity and visibility prediction ($tasks_{6-10}$).

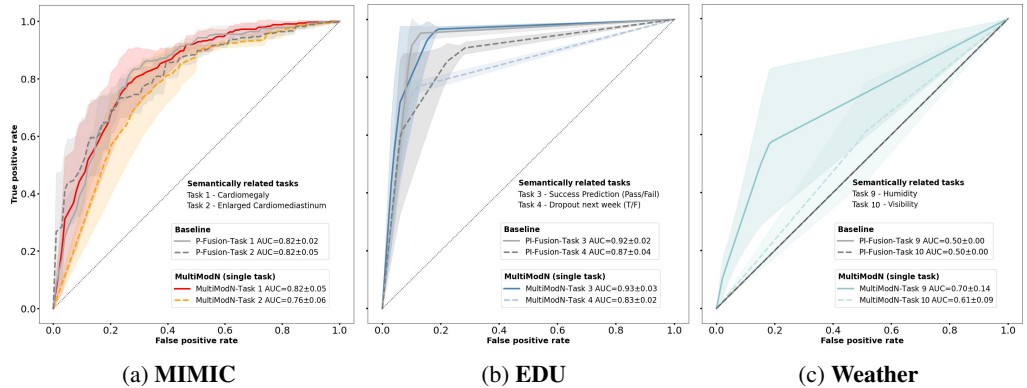

Figure 2: **MultiModN does not compromise performance in single-tasks.** AUROC for six binary prediction tasks in (a) MIMIC, (b) EDU, and (c) Weather2k. Tasks predicted by `P-Fusion` are compared with `MultiModN`. 95% CIs are shaded.

## 6 Experiments

**Overview.** We align feature extraction pipelines between `MultiModN` and the `P-Fusion` baseline in order to isolate the impact of parallel-monolithic vs. sequential-modular fusion (described in 4.1 and depicted in Appendix Sec. B). We thus do not expect a significant difference in performance, but rather aim to showcase the distinct benefits that can be achieved with modular sequential multimodal fusion *without compromising baseline performance*. In the following subsections, we perform four experiments to show these advantages. **[1]** `MultiModN` performance is not compromised compared to `P-Fusion` in single-task predictions. **[2]** `MultiModN` is able to extend to multiple tasks, also without compromising performance. **[3]** `MultiModN` is inherently composable and interpretable, providing modality-specific predictive feedback. **[4]** `MultiModN` is resistant to MNAR bias and avoids catastrophic failure when missingness patterns are different between train and test settings.

**Model evaluation and metrics.** All results represent a distribution of performance estimates on a model trained 5 times with different random weight initializations for the state vector and weights. Each estimate uses a completely independent test set from an 80-10-10 K-Fold train-test-validation split, stratified on one or more of the prediction targets. We report metrics (macro AUC, BAC, MSE) with 95% confidence intervals, as aligned with domain-specific literature of each dataset [34, 20, 36].

**Hyperparameter selection.** Model architectures were selected among the following hyperparameters: state representation sizes [1, 5, 10, 20, 50, 100], batch sizes [8, 16, 32, 64, 128], hidden features [16, 32, 64, 128], dropout [0, 0.1, 0.2, 0.3], and attention [0, 1]. These values were grouped into 3 categories (small, medium, large). We vary one while keeping the others fixed (within groups). Appendix Figure 9 shows that `MultiModN` is robust to changing batch size, while dropout rate and hidden layers negatively impact larger models (possibly overfitting). The most specific parameter to `MultiModN` is state size. As expected, we see negative impacts at size extremes, where small states likely struggle to transfer features between steps, while larger ones would be prone to overfitting.

### 6.1 Exp. 1: Sequential modularization in `MultiModN` does not compromise performance

**Setup.** A single-task model was created for each $task_{1-10}$ across all three datasets. Each model takes all modalities as input. We compare `MultiModN` and `P-Fusion` in terms of performance. AUROCs can be visualized in Figure 2 while BAC and MSE are detailed in Table 1. As feature extraction pipelines between `MultiModN` and `P-Fusion` are aligned, this experiment seeks to investigate if sequential modular fusion compromises model performance. To compress the multiple predictions of time-series into a single binary class, we select a representative time step (EDU $tasks_{3-4}$ at 60% course completion) or average over all time steps (Weather $tasks_{9-10}$ evaluated on a 24h window).

**Results.** Both `MultiModN` and `P-Fusion` achieve state-of-the-art results on single tasks using multimodal inputs across all 10 targets. In Figure 2c, we binarize the continuous weather task (humidity prediction) as an average across all time steps. The task is particularly challenging for the `P-Fusion` baseline, which has random performance (AUROC: 0.5). Compared with `P-Fusion`,

|  | MIMIC | | Education (EDU) | | | Weather | | | | |
|---|---|---|---|---|---|---|---|---|---|---|
|  | Cardiomegaly | ECM | Success | Dropout | Next Week | Temp. (24h) | Temp (72h) | Temp (720h) | Humidity | Visibility |
| *Metric* | *BAC* | *BAC* | *BAC* | *BAC* | *MSE* | *MSE* | *MSE* | *MSE* | *MSE* | *MSE* |
| **MultiModN** | 0.75 ±0.04 | 0.71 ±0.03 | 0.93 ±0.04 | 0.83 ±0.02 | 0.01 ±0.01 | 0.03 ±0.01 | 0.03 ±0.01 | 0.03 ±0.01 | 0.02 ±0.01 | 0.10 ±0.01 |
| **P-Fusion** | 0.75 ±0.02 | 0.69 ±0.03 | 0.92 ±0.03 | 0.87 ±0.05 | 0.01 ±0.01 | 0.02 ±0.01 | 0.03 ±0.01 | 0.02 ±0.01 | 0.03 ±0.01 | 0.08 ±0.02 |

Table 1: `MultiModN` **does not compromise performance in single-tasks.** Performance for binary and continuous prediction tasks in MIMIC, EDU, and Weather, comparing `P-Fusion` and `MultiModN`. 95% CIs are shown. *ECM: Enlarged Cardiomediastinum, Temp: Temperature*.

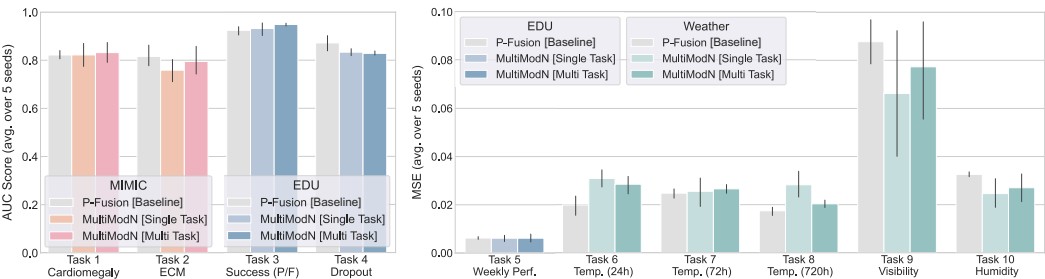

Figure 3: **Multi-task `MultiModN` maintains baseline performance in individual tasks.** Single- and multi-task `MultiModN` on the prediction of individual tasks, compared with the monolithic `P-Fusion` (can only be single-task). AUC for binary (**left**) and MSE for continuous **(right)**. Error bars: 95% CIs.

`MultiModN` shows a 20% improvement, which is significant at the $p < 0.05$ level. As the temporality of this task is particularly important, it could be hypothesized that the sequential nature of `MultiModN` better represents time-series inputs. Nevertheless, all weather targets are designed as regression tasks and show state-of-the-art MSE scores in Table 1 where `MultiModN` achieves baseline performance.

We provide an additional parallel fusion transformer baseline with experimental results showcased in Appendix Sec. E.4. The results indicate that `MultiModN` matches or outperforms the multimodal transformer in the vast majority of single- and multi-task settings, and comes with several inter­pretability, missingness, and modularity advantages. Specifically, using the primary metric for each task (BAC for classification and MSE for regression tasks), `MultiModN` beats the transformer baseline significantly in 7 tasks, overlaps 95% CIs in 11 tasks, and loses slightly (0.01) in 2 regression tasks.

> `MultiModN` matches `P-Fusion` performance across all 10 tasks in all metrics reported across all three multimodal datasets. Thus, modularity does not compromise predictive performance.

## 6.2 Exp. 2: Multi-task `MultiModN` maintains baseline performance in individual tasks

**Setup.** The modular design of `MultiModN` allows it to train multiple task-specific decoders and deploy them in any order or combination. While multi-task models have the potential to enrich feature extraction (and improve the model), it is critical to note that all feature extraction from the raw input is performed before `MultiModN` is trained. `MultiModN` is trained on embeddings extracted from pre­trained models (independently of its own encoders). This is done purposely to best isolate the effect of parallel-monolithic vs. sequential-modular fusion. We train three multi-task `MultiModN` models (one for each dataset, predicting the set of tasks in that dataset, i.e. $tasks_{1-2}$ in MIMIC, $tasks_{3-5}$ in EDU, and $tasks_{6-10}$ in Weather) and compare this to 10 single-task `MultiModN` models (one for each $tasks_{1-10}$). Monolithic models, like `P-Fusion` are not naturally extensible to multi-task predictions. Thus `P-Fusion` (grey bars in Figure 3) can only be displayed for single-task models. This experiment aims to compare `MultiModN` performance between single- and multi-task architectures to ensure that this implementation does not come at a cost to the predictive performance of individual tasks.

**Results.** In Figure 3 we compare the single-task `P-Fusion` (grey bars), to single- and multi­task implementations of `MultiModN` (in color). The results demonstrate that `MultiModN` is able to maintain its performance across all single-prediction tasks even when trained on multiple tasks. We additionally include the results of our model on various numbers and combinations of inputs, described further in Appendix Sec. E.5. The baseline would have to impute missing features in these combinations, exposing it to catastrophic failure in the event of systematic missingness (Sec. 6.4).

> MultiModN has the significant advantage of being naturally extensible to the prediction of multiple tasks without negatively impacting the performance of individual tasks.

## 6.3 Exp. 3: MultiModN has inherent modality-specific local and global model explainabilty

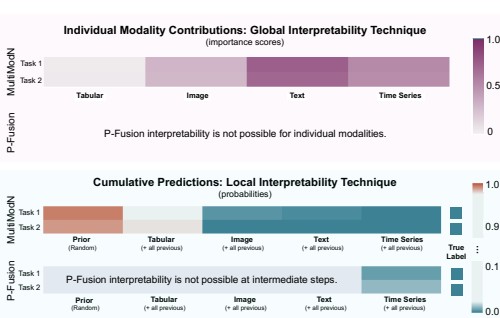

Figure 4: **Inherent modality-specific model explainability in** MultiModN. Heatmaps show individual modality contributions (IMC) **(top)** and cumulative contributions (CP) **(bottom)**: respectively **importance score** (global explainability) or **cumulative probability** (local explainability). The multi-task MultiModN for $task_{1-2}$ in MIMIC is compared to two single-task P-Fusion models. IMC are only possible for MultiModN (only 1 modality encoded, rest are skipped). CP are made sequentially from states encoding all previous modalities. P-Fusion is unable to naturally decompose modality-specific contributions (can only make predictions once all modalities are encoded). IMC is computed across all patients in the test set. CP is computed for a single patient, (true label = 0 for both $task_{1-2}$). The CP heatmap shows probability ranging from **confident negative diagnosis (0)** to **perfect uncertainty** and **confident positive diagnosis (1)**.

**Setup.** Parallel MM fusion obfuscates the contribution of individual inputs and requires add-on or post hoc methods to reveal unimodal contributions and cross-modal interactions [37, 38, 39]. Soenksen et al. [20] used Shapley values [40] to derive marginal modality contributions. While these post hoc methods provide valuable insight, they are computationally expensive and challenging or impossible to deploy at inference. In contrast, MultiModN confers inherent modality-specific interpretability, where the contribution of each input can be decomposed by module. We use $task_{1-2}$ in MIMIC to compute two measures: **[1] Importance score**, where each encoder is deployed alone, providing predictive importance of a single modality by subtracting predictions made from the prior state. This can be computed across all data points or individual data points. **[2] Cumulative probability**, where the prediction from each multi-task decoder is reported in sequence (i.e. given the previously encoded modalities). We demonstrate this on a random patient from the test set, who has a true label of 0 for both tasks. Further plots are in Appendix Sec. E.2.

**Results.** Monolithic P-Fusion models cannot be decomposed into any modality-specific predictions, and its (single-task) prediction is only made after inputting all modalities. In contrast, Figure 4 shows MultiModN provides granular insights for both importance score and cumulative prediction. We observe that the Text modality is the most important. The cumulative prediction shows the prior strongly predicts positivity in both classes and thus that $S_0$ has learned the label prevalence.

> The predictions naturally produced by MultiModN provide diverse and granular interpretations.

## 6.4 Exp. 4: MultiModN is robust to catastrophic failure from biased missingness

**Setup.** MultiModN is designed to predict any number or combination of tasks from any number or combination of modalities. A missing modality is skipped (encoder $e_i$ is not used) and not padded/encoded. Thus, MultiModN avoids featurizing missingness, which is particularly advantageous when missingness is MNAR. Featurizing MNAR can result in catastrophic failure when MNAR patterns differ between train and test settings. We demonstrate MultiModN's inherent robustness to catastrophic MNAR failure by training MultiModN and P-Fusion on four versions of MIMIC with various amounts (0, 10, 50, or 80%) of MNAR by artificially removing one modality in one class only. Figure 5 compares MultiModN and P-Fusion on $task_1$ when tested in a setting that has either no missingess or where the MNAR pattern is different (i.e. label-flipped).

**Results.** Figure 5 shows a dramatic catastrophic failure of P-Fusion in a label-flipped MNAR test set (**black solid line**) compared with MultiModN. P-Fusion is worse than random at 80% MNAR (AUROC: 0.385). In contrast, MultiModN only loses 10% in MNAR flip, remarkably, matching performance in a test with no missingness. Further plots in Appendix E.3.

`MultiModN` is robust to catastrophic missingness (MNAR failure) where `P-Fusion` is not.

## 7  Conclusion

We present `MultiModN`, a novel sequential modular multimodal (MM) architecture, and demonstrate its distinct advantages over traditional monolithic MM models which process inputs in parallel.

By aligning the feature extraction pipelines between `MultiModN` and its baseline `P-Fusion`, we better isolate the comparison between modular-sequential MM fusion vs. monolithic-parallel MM fusion. We perform four experiments across 10 complex real-world MM tasks in three distinct domains. We show that neither the sequential modularization of `MultiModN` nor its extension to multi-task predictions compromise the predictive performance on individual tasks compared with the monolithic baseline implementation. Training a multi-task model can be challenging to parameterize across inter- and cross-task performance [13, 41]. We perform no specific calibration and show that `MultiModN` is robust to cross-task bias. Thus, at no performance cost, modularization allows the inherent benefits of multi-task modeling, as well as providing interepretable insights into the predictive potential of each modality. The most significant benefit of `MultiModN` is its natural robustness to catastrophic failure due to differences in missingness between train and test settings. This is a frequent and fundamental flaw of many domains and especially impacts low-resource settings where modalities may be

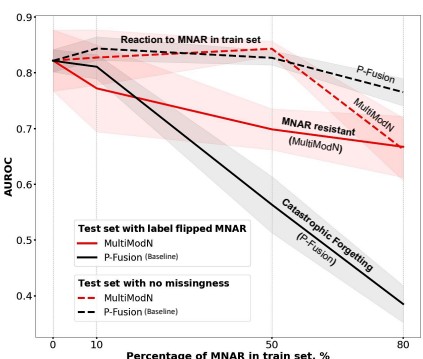

Figure 5: `MultiModN` **is robust to catastrophic MNAR failure.** Impact of MNAR missingess on `MultiModN` vs. `P-Fusion`. Both models are trained on four versions of the MIMIC dataset with 0—80% MNAR. They are then tested on either a test set with no MNAR missingness (- - -) or a test set where the biased missingness is label-flipped, i.e. MNAR occurs in the other binary class as compared with the train (—). Results for $task_1$ depicted. CI95% shaded.

missing for reasons independent of the missingness in the train set. More generally, modularization creates a set of self-contained modules, composable in any number or combination according to available inputs and desired outputs. This composability not only provides enormous flexibility at inference but also reduces the computational cost of deployment. Taken together, these features allow `MultiModN` to make resource-adapted predictions, which have a particular advantage for real-world problems in resource-limited settings.

**Limitations and future work.** The main limitation for studying MM modeling is the scarcity of large-scale, open-source, MM datasets that cover multiple real-world tasks, especially for time-series. Additionally, while `MultiModN` is theoretically able to handle any number or combination of modalities and tasks, this has not been empirically tested. Having a high combinatorial generalization comes at a computational and performance cost, where the 'memory' of a fixed-size state representation will likely saturate at scale. The performance of `MultiModN` is purposely limited in this work by fixing the feature extraction pipeline, to best isolate the effect of sequential fusion. Future work leveraging `MultiModN` model-agnostic properties would be able to explore the potential performance benefit. This is particularly interesting for time-series, for which the state 'memory' may need to be parameterized to capture predictive trends of varying shapes and lengths.

## 8  Acknowledgements

This project was substantially co-financed by the Swiss State Secretariat for Education, Research and Innovation (SERI).

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

# A  `MultiModN` Framework Implementation

A **task- and modality-agnostic open-source framework** `MultiModN` solution has been implemented in Python using PyTorch as the primary machine learning framework. The `/multimodn` package contains the `MultiModN` model and its components. The `/datasets` package is responsible of preparing the data inputs for `MultiModN`. Some examples using the public Titanic dataset have been provided.

The code is available at: `https://github.com/epfl-iglobalhealth/MultiModN`.

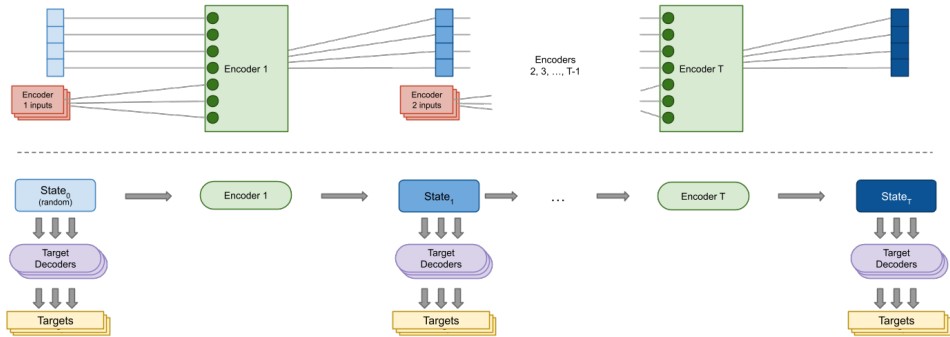

Figure 6: **Architecture of `MultiModN` code**. The **upper panel** shows a more detailed depiction of sequential encoding using a series of model-agnostic encoders which receive inputs of variable dimension to create the evolving state vector, which represents the shared feature space. The **lower panel** shows how each state can be probed by any number of target decoders.

## A.1  `MultiModN` metrics

During training and evaluation, the metrics of the model are stored in a log at each epoch in a matrix of dimensions $(E + 1) * D$, where $E$ is the number of encoders and $D$ the number of decoders. Each row represents the metrics for a target at each state of the model.

## A.2  Code structure

### A.2.1  MultiModN

`/multimodn` package contains the `MultiModN` model and its modules:

**[1]** Encoders: `/multimodn/encoders`
**[2]** Decoders: `/multimodn/decoders`
**[3]** State: `/multimodn/state.py`

### A.2.2  Datasets

`/dataset` package contains the MultiModDataset abstract class, compatible with `MultiModN`.

Specific **datasets** are added in the `/dataset` directory and must fulfill the following requirements:

- Contain a dataset class that inherit MultiModDataset or has a method to convert into a MultiModDataset
- Contain a `.sh` script responsible of getting the data and store it in `/data` folder

`__getitem__` function of MultiModDataset subclasses must yield elements of the following shape:

```
tuple
(
    data: [torch.Tensor],
    targets: numpy.ndarray,
```

```
    (optional) encoding_sequence: numpy.ndarray
)
```

namely a tuple containing an array of tensors representing the features for each subsequent encoder, a numpy array representing the different targets and optionally a numpy array giving the order in which to apply the encoders to the subsequent data tensors. Note: `data` and `encoding_sequence` must have the same length.

**Missing values.** The user is able to choose to keep missing values (`nan` values). Missing values can be present in the tensors yielded by the dataset and are managed by `MultiModN`.

### A.2.3 Pipelines

`/pipeline` package contains the training pipelines using `MultiModN` for Multimodal Learning. It follows the following steps:

- Create `MultiModDataset` and the `dataloader` associated
- Create the list of encoders according to the features shape of the MultiModDataset
- Create the list of decoders according to the targets of the MultiModDataset
- Init, train and test the `MultiModN` model
- Store the trained model, training history and save learning curves

### A.3 Quick start

Quick start running `MultiModN` on Titanic example pipeline with a Multilayer Perceptron encoder:

```
./datasets/titanic/get_data.sh
python3 pipelines/titanic/titanic_mlp_pipeline.py
```

Open `pipelines/titanic/titanic_mlp.png` to look at the training curves.

## B Additional details about `MultiModN` Architecture

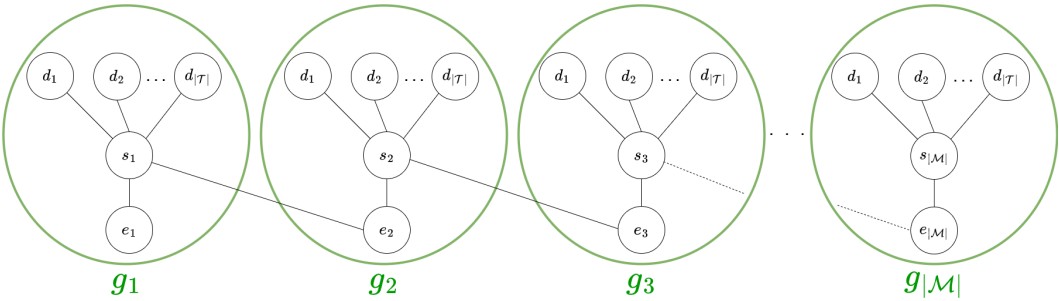

Figure 7: **Schematic representation of the *modules* ($g$, groups) of** `MultiModN`. $e$: encoders, $s$: state vector, $d$: decoders. Each module is connected by a single edge between $s_n$ and $e_{n+1}$. There are $|\mathcal{M}|$ groups (i.e. input-specific modules) and $|\mathcal{T}|$ decoders per module.

**Modularity.** In the following, we detail the computation of `MultiModN`'s modularity measure. The total number of edges in a `MultiModN` module is $|\mathcal{T}| + 1$. The total number of modules is $|\mathcal{M}|$ and there are $|\mathcal{M}| - 1$ edges connecting consecutive modules, which makes for a total number of edges in the entire `MultiModN` model of $m = |\mathcal{M}|(|\mathcal{T}| + 2) - 1$.

To compute the modularity following using the formalization proposed by Newman et al [29], we need to define groups. In the case of `MultiModN`, each group corresponds to one *module*. Let $G$ be the matrix whose components $g_{ij}$ is the fraction of edges in the original network that connect vertices in group $i$ to those in group $j$.

Within `MultiModN`, each group contains $(|\mathcal{T}| + 1)$ edges and is connected to adjacent groups by two edges, with the exception of $g_1$ and $g_{|\mathcal{M}|}$, which are connected to only one other group (cf. Figure 7).

Thus, $G$ is a tridiagonal matrix whose diagonal elements $G_{ii}$ are equal to $(|\mathcal{T}| + 1)/m$ and whose upper and lower diagonal elements are equal to $1/m$:

$$
G = \frac{1}{m}
\begin{pmatrix}
|\mathcal{T}|+1 & 1 & & & & \\
1 & |\mathcal{T}|+1 & 1 & & & \\
& \ddots & & \ddots & & \ddots \\
& & 1 & & |\mathcal{T}|+1 & 1 \\
& & & & 1 & |\mathcal{T}|+1
\end{pmatrix}
$$

The modularity measure is defined by $\mathcal{Q} = \mathrm{tr}(G) - \|G^2\|$, where $\mathrm{tr}(G)$, is the trace of $G$ and $\|G^2\|$ is the sum of elements of $G^2$. The trace of $G$ is equal to $\frac{|\mathcal{M}|}{m}(\mathcal{T} + 1)$ and we have $G^2$ equal to:

$$
\frac{1}{m^2}
\begin{pmatrix}
(|\mathcal{T}|+1)^2+1 & 2(|\mathcal{T}|+1) & 1 & & & \\
2(|\mathcal{T}|+1) & (|\mathcal{T}|+1)^2+2 & 2(|\mathcal{T}|+1) & 1 & & \\
1 & 2(|\mathcal{T}|+1) & (|\mathcal{T}|+1)^2+2 & 2(|\mathcal{T}|+1) & 1 & \\
& \ddots & \ddots & \ddots & \ddots & \ddots \\
& & 1 & 2(|\mathcal{T}|+1) & (|\mathcal{T}|+1)^2+2 & 2(|\mathcal{T}|+1) \\
& & & 1 & 2(|\mathcal{T}|+1) & (|\mathcal{T}|+1)^2+1
\end{pmatrix}
$$

Hence, we have:

$$
\begin{aligned}
\|G^2\| &= \frac{1}{m^2}\left[|\mathcal{M}|((|\mathcal{T}|+1)^2+2) - 2 + 2(|\mathcal{M}|-1)2(|\mathcal{T}|+1) + 2(|\mathcal{M}|-2)\right] \\
&= \frac{|\mathcal{M}||\mathcal{T}|^2 + 6|\mathcal{M}||\mathcal{T}| + 9|\mathcal{M}| - 4|\mathcal{T}| - 10}{|\mathcal{M}|^2|\mathcal{T}|^2 + 4|\mathcal{M}|^2|\mathcal{T}| + 4|\mathcal{M}|^2 - 2|\mathcal{M}||\mathcal{T}| - 4|\mathcal{M}| + 1}
\end{aligned}
$$

It is important to note that when $\mathcal{M}$ increases, the trace will tend to $(|\mathcal{T}| + 1)/(|\mathcal{T}| + 2) \approx 1$ for a large number of tasks. Moreover, when $|\mathcal{M}|$ increases $\|G^2\|$ will decrease towards to 0. Thus, for a large number of tasks, we have a modularity measure that increases and tends to 1 with the number of modalities.

**Architecture alignment between** `P-Fusion` **and** `MultiModN`**.** We purposely align the feature extraction pipelines of `P-Fusion` and `MultiModN` in order to best isolate the effect of monolithic-parallel fusion vs. modular-sequential fusion. In Figure 8, we see how the alignment is limited to the input, where both `MultiModN` and `P-Fusion` share the feature-extraction of each modality, where the `MultiModN` encoders receive an embedding ($emb$). No element of `MultiModN` is changed as described in Figure 1. Embeddings from missing data can be skipped (i.e. not encoded) by `MultiModN`.

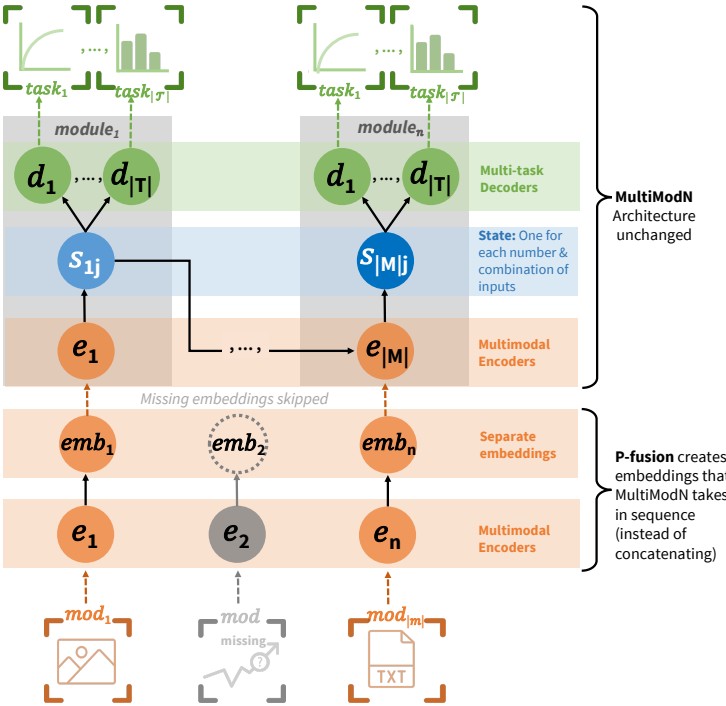

Figure 8: **Alignment of** `P-Fusion` **and** `MultiModN` **architectures.** We purposely ensure that the feature extraction pipelines are aligned between `P-Fusion` and `MultiModN`. To this end, we use the embeddings ($emb$) produced by `P-Fusion` as inputs into the `MultiModN` encoders ($e$). No element of `MultiModN` is changed. `MultiModN` encoders ($e$) are in orange, the `MultiModN` state ($s$) is in blue and multi-task `MultiModN` decoders ($d$) are in green.

For MIMIC, feature extraction for each modularity is replicated from previous work [20] and we use embeddings generated from a set of pre-trained models. For Weather and Education, as there were no pre-existing embedding models, we design autoencoders trained to reconstruct the original input features from a latent space. We keep the autoencoder's encoder and decoder structure exactly aligned with the encoders of `MultiModN` (two ReLU activated, fully-connected Dense layers and a third layer either generating the state representation with a ReLU activation or the final prediction with a sigmoid activation). We also align the number of trainable parameters with `MultiModN`'s modality-based encoders for a fair baseline comparison by selecting an appropriate state representation size per modality to equal the state representation in `MultiModN`. The remaining hyperparameters are left exactly the same (batch size, hidden layer size, dropout rate, optimization metrics, loss function).

## C  Datasets and Tasks

A description of all tasks is provided in Table 2. In the following paragraphs, we detail the preprocessing decisions on the datasets for context and reproducibility.

**MIMIC.** The data includes four input modalities (tabular, textual, images, and time-series) derived from several sources for each patient. We align our preprocessing pipeline exactly with the study from which our baseline of `P-Fusion` is derived [20] (described in 4.1). To this end, we use patient-level feature embeddings extracted by the pre-trained models described in [20] and depicted in Figure 8.

The dataset is a combination of two MIMIC databases: MIMIC-IV v.1.0 [21] and MIMIC-CXR-JPG v.2.0.0 [22]. After gaining authorized access to PhysioNet online repository [30], the embedding dataset can be downloaded via this link: `https://physionet.org/content/haim-multimodal/1.0.1/`. The dataset comprises $45,050$ samples, each corresponding to a time point during a patient's hospital stay when a chest X-ray was obtained. It covers a total of $8,655$ unique patient stays. To ensure data quality and limit our experiments to two thematic tasks (diagnosis of $task_1$: cardiomegaly and $task_2$: enlarged cardiomediastinum), we remove duplicates (based on image id and image acquisition time), and retain only relevant samples that have valid labels for both targets of interest,

i.e. both $task_1$ and $task_2$ are either present (1) or absent (0). Subsequent experiments for these tasks are thus performed on the $921/45,050$ selected relevant patients.

**EDU.** This dataset involves hand-crafted features extracted for $5,611$ students across 10 weeks of data. The preprocessing of data is an exact replication of several related works using the same dataset [34, 31, 42] based on 4 feature sets determined as predictive for MOOC courses in [43]. 45 features regarding problem and video data are extracted per student per week, covering features like *Delay Lecture*, which calculates the average delay in viewing video lectures after they are released to students or *TotalClicksProblem*, the number of clicks that a student has made on problems this week. The features are normalized with min-max normalization, and missing values are imputed with zeros that have meaning i.e. no problem events in a week is correctly inferred as zero problems attempted. In this setting, missingness is a valued, predictive feature of the outcome, and thus we do not perform missingness experiments on this dataset. MultiModN has the ability to select whether missingenss is encoded or not, and thus it would not suffer a disadvantage in a setting where missingness should be featurized.

In MOOCs, a common issue is that students join a course and never participate in any assignments, homeworks, or quizzes. This could be due to registering aspirationally, to read some material, or to watch videos [44]. An instructor can easily classify students who have never completed an assignment as failing students. As introduced by Swamy et al. in [34] and used with this dataset in related work [31, 42, 33], EDU has removed students that were predicted to fail in the first two weeks simply by having turned in no assignments (99% confidence of failing with an out-of-the-box logistic regression model, where the confidence threshold was tuned over balanced accuracy calculations). It has been shown that including these students will artificially increase the performance of the model, providing even better results than those showcased by MultiModN in this work [34]. We thus exclude these students to test a more challenging modeling problem.

**Weather.** The Weather2k dataset, presented in [36], covers features from $1,866$ weather stations with 23 features covering seven different units of measurements (degrees, meters, HPA, celsius, percentage, $ms^{-1}$, millimeters). To align these features on vastly different scales, we normalize the data. We use the large extract (R) provided by the authors instead of the smaller representative sample also highlighted in the benchmark paper (S) [36]. We use the first 24 hourly measurements as input to train the MultiModN model and calculate the five regression tasks as determined in Table 2 below.

**Tasks.** As showcased in Table 2, our evaluation covers 10 binary and regression tasks in two settings: static (one value per datapoint) or continuous (changing values per datapoint, per timestep).

| | Task | Type | Name | Description |
|---|---|---|---|---|
| **MIMIC** | 1 | Static Binary | Cardiomegaly | Labels determined as per [20] using NegBio [45] and CheXpert [46] to process radiology notes, resulting in four diagnostic outcomes: positive, negative, uncertain, or missing. |
| | 2 | Static Binary | Enlarged Cardiomediastatinum | Labels determined as per [20] using NegBio [45] and CheXpert [46]. The set of label values is identical to the one for cardiomegaly. |
| **EDU** | 3 | Static Binary | Student Success Prediction | End of course pass-fail prediction (per student) as per [34] on the course. |
| | 4 | Continuous Binary | Student Dropout Prediction | 1 if student has any non-zero value on a video or problem feature from next week until the end of the course, 0 if not. Not valid for the last week, so the task involves n-1 decoder steps for n timesteps. Can be easily extended to a multiclass task by separating video or problem involvement until the end of the course into separate classes. |
| | 5 | Continuous Regression | Next Week Performance Forecasting | Moving average (per student, per week) of three student performance features from [43] and removed in baseline paper [34]: *Student Shape* (recieving the maximum quiz grade on the first attempt), *CompetencyAlignment* (number of problems the student has passed this week), *CompetencyStrength* (extent to which a student passes a quiz getting the maximum grade with few attempts). |
| **Weather** | 6 | Continuous Regression | Short Term Temperature Forecasting | Changing air temperature measurements (collected per station, per hour), shifted by 24 hourly measurements (1 day). |
| | 7 | Continuous Regression | Mid Term Temperature Forecasting | Changing air temperature measurements (collected per station, per hour), shifted by 72 hourly measurements (3 days). |
| | 8 | Continuous Regression | Long Term Temperature Forecasting | Changing air temperature measurements (collected per station, per hour), shifted by 720 hourly measurements (30 days). |
| | 9 | Static Regression | Relative Humidity | Instantaneous humidity relative to saturation as a percentage at 48h from 2.5 meters above the ground, as used as a benchmark forecasting task in [36]. |
| | 10 | Static Regression | Visibility | 10 minute mean horizontal visibility in meters at 48 hr from 2.8 meters above the ground, as used as a benchmark forecasting task in [36] |

Table 2: Description of all 10 tasks used to evaluate MultiModN.

Note that for $task_4$, the three features used to calculate next week's performance were not included in the original input features because of possible data leakage, as student performance on quizzes directly contributes to their overall grade (Pass/Fail).

For the AUROC curves on $tasks_{9-10}$ in Figure 2, we conduct a binarization of the two last regression tasks. To align with the regression task, we conduct a static forecasting prediction per station for the relative humidity or visibility with a time window of 24h for the 48th timestep (one day in advance). While `MultiModN` is capable of making a prediction at each continuous timestep, `P-Fusion` is not able to do this without a separate decoder at each timestep, and therefore to compare the tasks we must choose a static analysis. We choose a threshold based on the normalized targets: 0.75 for humidity and 0.25 for visibility (selected based on the distribution of the feature values for the first 1000 timesteps), and evaluate the predictions as a binary task over this threshold.

Analogously, in the Section E analysis below on the binarization of the remaining regression tasks, we express the continuous regression tasks for temperature forecasting $tasks_{6-8}$ as a static binary task. To do this, we evaluate the prediction from the full window (24th timestep) at the respective forecasting timestep (48, 96, 744). The threshold we select is 0.3, closely corresponding to the normalized mean of the temperature (0.301) over the first 1000 timesteps.

## D    Model Optimization

Table 3 indicates the chosen hyperparameters for the experiments conducted in Section 6 of the paper, selected based on the optimal hyperparameters for the multi-task settings (all the tasks for a dataset predicted jointly). The single tasks use the same hyperparameters as the multi-task settings. For saving the best model across training epochs in the time series settings (EDU, Weather), our optimization metric saved the ones with best validation set results on the most short term task. Therefore, for EDU, we use MSE of $task_5$, predicting next week performance, and $task_6$ for Weather, forecasting temperature within a day). The intuition is that choosing the best model on the short term task would allow the model to emphasize stronger short-term connections, which in turn would improve long term performance.

| Task | # of Timesteps | Batch Size | Dropout Rate | Hidden Layer Size | State Rep. Size | Save Best Model (chosen metric) |
|---|---|---|---|---|---|---|
| 1 | 1 | 16 | 0.2 | 32 | 50 | $task_1$ Val BAC + Macro AUROC |
| 2 | 1 | 16 | 0.2 | 32 | 50 | $task_2$ Val BAC + Macro AUROC |
| **MIMIC** | **1** | **16** | **0.2** | **32** | **50** | $tasks_{1-2}$ **BAC + Macro AUROC** |
| 3 | 10 | 64 | 0.1 | 32 | 20 | $task_3$ Val Accuracy |
| 4 | 10 | 64 | 0.1 | 32 | 20 | $task_4$ Val Accuracy |
| 5 | 10 | 64 | 0.1 | 32 | 20 | $task_5$ Val MSE |
| **EDU** | **10** | **64** | **0.1** | **32** | **20** | $task_5$ **Val MSE** |
| 6 | 24 | 128 | 0.1 | 32 | 20 | $task_6$ Val MSE |
| 7 | 24 | 128 | 0.1 | 32 | 20 | $task_7$ Val MSE |
| 8 | 24 | 128 | 0.1 | 32 | 20 | $task_8$ Val MSE |
| 9 | 24 | 128 | 0.1 | 32 | 20 | $task_9$ Val MSE |
| 10 | 24 | 128 | 0.1 | 32 | 20 | $task_{10}$ Val MSE |
| **Weather** | **24** | **128** | **0.1** | **32** | **20** | $task_6$ **Val MSE** |

Table 3: Hyperparameters selected for each experiment. We tuned the hyperparameters for the multi-task models (MIMIC, EDU, Weather) and used the same hyperparameters for each single-task model for a fair comparison.

In Figure B, we examine a case study of `MultiModN`'s changing performance on $task_3$, student success prediction in the EDU dataset, by varying hyperparameters across three model architectures (chosen for small, medium, and large hyperparameter initializations). We note that batch size is fairly robust across all three initial model settings, with large batch size on the largest model having slightly variable performance. Examining changing dropout rate, we note that with medium and large models, change in dropout impacts performance considerably. This allows us to hypothesize that high dropout on larger state representations does not allow the model to learn everything it can from the data.

Looking at varied hidden layer size, we see comparable performance for the small and medium initializations, but note that in the large case, having a smaller hidden layer size is important to maintain performance. Even as `MultiModN` performance trends upwards with larger hidden layer size (i.e. 128) for the large initialization, the confidence interval is large, so performance is not stable. Lastly, observing state representation size, we see that when state representation is too small for the task (i.e. 1, 5), the small and large models are adversly impacted. Additionally, when state representation is too large (i.e. 100), performance seems to drop or increase variability again. It is therefore important to tune `MultiModN` and find the right state representation size for the dataset and predictive task(s).

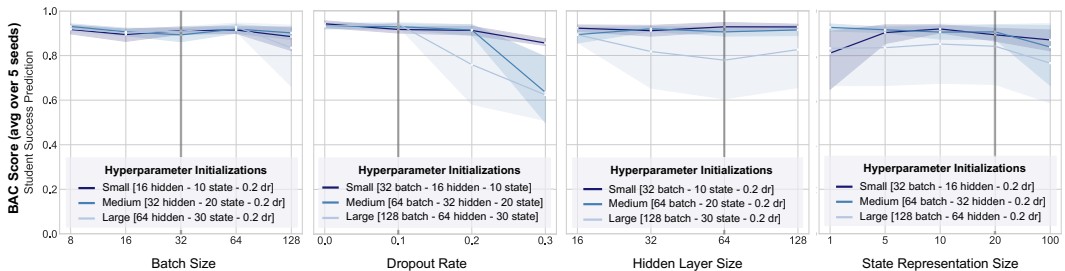

Figure 9: `MultiModN` hyperparameter selection across four parameters on $task_3$ of the EDU dataset (Pass/Fail). Each individual parameter is varied on the x-axis (***dr: dropout rate***) with all other initializations fixed (grouped in small, medium, and large values). These are compared in terms of balanced accuracy (BAC). 95% CIs are shaded.

**Experimental Setup.**     For the results reported in Sections 6.1, 6.2 and 6.4, we perform 5-fold stratified cross-validation with 80-10-10 train-validation-test split. Due to the time-series nature of the EDU and Weather datasets, we orient the stratification the real labels associated with the longest-term task. For EDU, $task_3$ (end of course pass-fail prediction) and for Weather $task_8$ (30-day temperature forecasting) is chosen for the stratification split.

In an alternative approach, regarding the MIMIC dataset, a two-step procedure was implemented to address the imbalanced class ratios, given the absence of a prioritized task. Initially, a new dummy label was assigned to each sample, indicating positivity if both pathologies are present and negativity otherwise. Subsequently, a label was assigned to each unique hospital stay based on the aggregated labels from the first step. A hospital stay was considered positive if the number of times sample from that stay has been found positive is greater than or equal to the half of samples with the same hospital stay ID. The latter, as outlined in [20], ensured that no information was leaked on the hospital stay level during stratification.

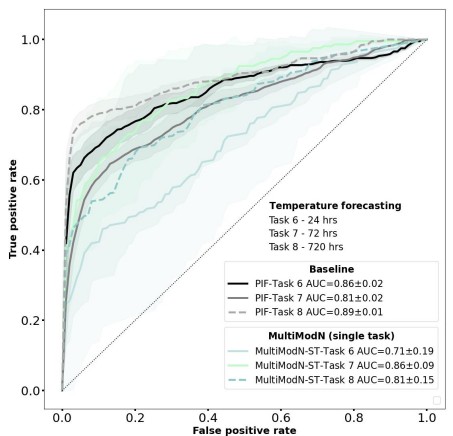

Figure 10: AUROC for three additional binary prediction tasks in Weather2k. Targets predicted by `P-Fusion` are compared to `MultiModN`. 95%CIs are shaded.

Experiments were conducted using the same architecture in PyTorch (MIMIC) and TensorFlow (EDU, Weather), to provide multiple implementations across training frameworks for ease of use. We use an Adam optimizer with gradient clipping across all experiments.

# E   Additional Experiments

## E.1   Single task

We present the binarized results (AUROC curves) for several additional regression tasks ($tasks_{6-8}$ for Weather) in Figure 10. The specific details of binarization are discussed above in Appendix

Section C. We note that the confidence intervals overlap for `P-Fusion` and `MultiModN` over all Weather tasks. However, it is clear that the performance of `MultiModN` varies a lot (large CIs). This could be due to the design of the binarization as originally in the benchmark paper [36], this was introduced as a regression task. Another contributing factor to large CIs could be that the model was trained across all timesteps but only evaluated on one timestep for a comparable binarization. Despite these caveats, we can statistically conclude that `P-Fusion` and `MultiModN` performance are comparable on these additional tasks.

### E.2 Interpretability

We perform an interpretability analysis for the EDU dataset, analogous to the local and global interpretability analysis on MIMIC from Figure 4. The global analysis (IMC) is conducted over all students for the first week of the course. We note interesting findings: specifically that problem interactions are more important for $tasks_{3-4}$ while video interactions are more important for $task_5$.

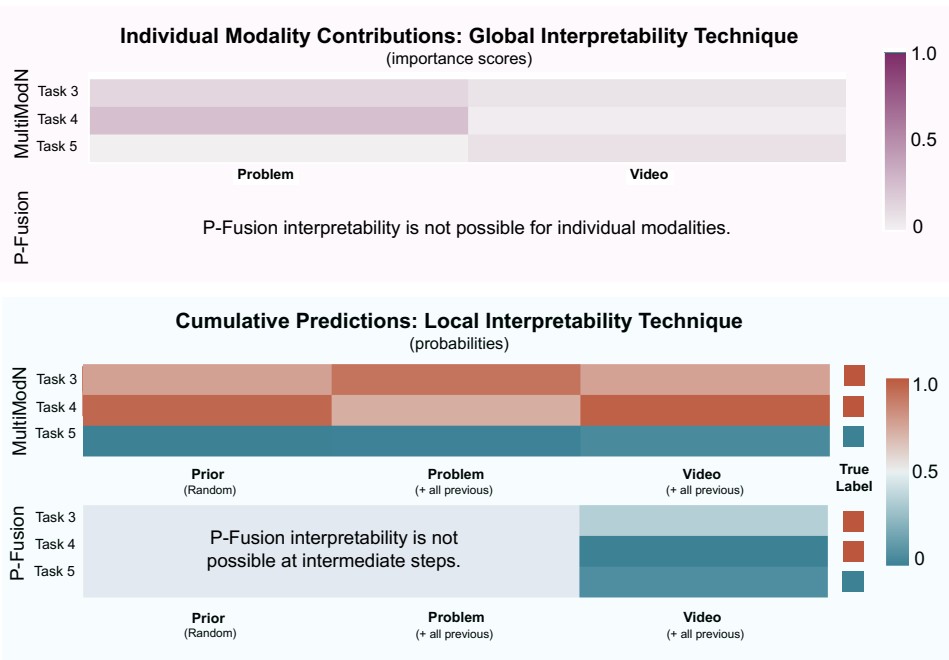

Figure 11: **Inherent modality-specific model explainability in** `MultiModN` **for** $tasks_{3-5}$**.** Heatmaps show individual modality contributions (IMC) **(top)** and cumulative contributions (CP) **(bottom)**: respectively **importance score** (global explainability) or **cumulative probability** (local explainability). The multi-task `MultiModN` for $task_{3-5}$ in EDU is compared to two single-task `P-Fusion` models. IMC are only possible for `MultiModN` (only 1 modality encoded, rest are skipped). CP are made sequentially from states encoding all previous modalities. `P-Fusion` is unable to naturally decompose modality-specific contributions (can only make predictions once all modalities are encoded). IMC is computed across all students in the test set. CP is computed for a single student, (true label = 1 for $tasks_{3-4}$ and 0 for $task_5$). The CP heatmap shows probability ranging from **confident negative diagnosis (0)** to **perfect uncertainty** and **confident positive diagnosis (1)** .

The student selected for the CP local analysis passes the course (1 for $task_3$) and does not dropout (1 for $task_4$), but does not have strong performance in the next week ($\sim 0$ in $task_5$). This analysis is also conducted across the first week of course interactions. We see that `P-Fusion` cannot produce modality-specific interpretations and predicts the incorrect label. However, `MultiModN` is able to identify a changing confidence level across modalities, eventually ending on the right prediction for all tasks. The confidence for $task_3$ increases with the student's problem interactions and reduces for their video interactions. This could have potential for designing an intervention to improve student learning outcomes. We note that for $task_5$, both the students' problem and video interactions contribute similarly to the prediction.

## E.3 Missingness

We expand on the missingness experiments presented in 6.4. Here, we present further control experiments (training on data missing-at-random, MAR) in both MIMIC tasks ($task_1$: diagnosis of Cardiomegaly 12 and $task_2$: diagnosis of Enlarged Cardiomediastinum 13).

In the first two subplots of each figure, both P-Fusion (black) and MultiModN (red) are trained on MNAR data and then evaluated either on a test set at risk of catastrophic failure (where the pattern of MNAR is label-flipped, first figure) or on a test set with no missingness. As can be seen in the first figures, P-Fusion suffers catastrophic failure in the MNAR flip, becoming worse than random when a single modality is missing at 80%, as opposed to MultiModN, which only decreases AUROC about 10%. When the test set has no missing values, P-Fusion and MultiModN are not significantly different, proving that the catastrophic failure of P-Fusion is due to MNAR. This is further confirmed in the last two plots of each task, where the models are trained on MAR data and evaluated on test sets either without missing values or MAR missingness.

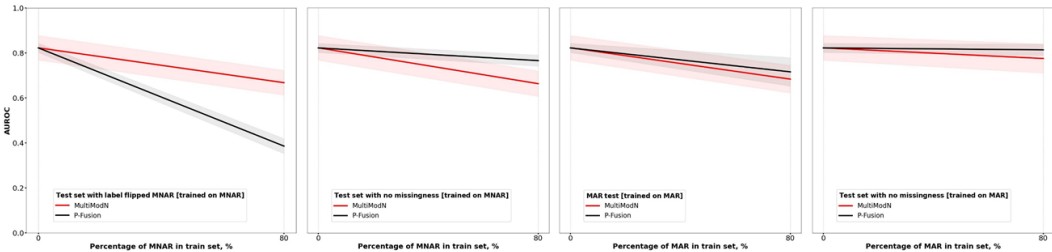

Figure 12: **Detailed missingness experiments for** $task_1$ **(Cardiomegaly).** P-Fusion (black) and MultiModN (red) are trained on MIMIC data where various percentages of a single modality are missing (0 or 80%) either for a single class (MNAR, first two plots) or without correlation to either class (MAR, last two plots). The AUROCs are shown for each when evaluated on test sets which either have a risk of catastrophic failure (first plot, MNAR with label flip) or on test sets without missingness or MAR missingness. CI95% shaded.

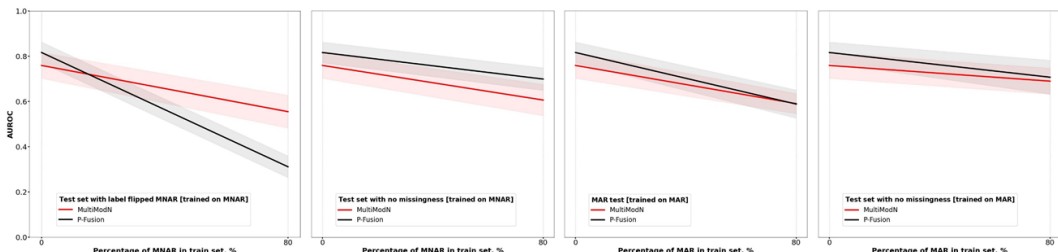

Figure 13: **Detailed missingness experiments for** $task_2$ **(Enlarged Cardiomediastinum).** P-Fusion (black) and MultiModN (red) are trained on MIMIC data where various percentages of a single modality are missing (0 or 80%) either for a single class (MNAR, first two plots) or without correlation to either class (MAR, last two plots). The AUROCs are shown for each when evaluated on test sets which either have a risk of catastrophic failure (first plot, MNAR with label flip) or on test sets without missingness or MAR missingness. CI95% shaded.

## E.4 Comparison to a P-Fusion Transformer

Additional experiments with a Transformer have been conducted on 10 tasks across three datasets. Results are showcased below in two tables (left for $tasks_{1-4}$ and right for $tasks_{5-10}$) with 95% CIs.

The hyperparameter-tuned architecture (based on head size, number of transformer blocks, MLP units) for EDU and Weather is a transformer model with 4 transformer blocks, 4 heads of size 256, dropout of 0.25, MLP units 128 with dropout 0.4, batch size 64, trained for 50 epochs with cross-entropy loss. For MIMIC, the most performant (tuned) transformer architecture includes 2 transformer blocks with 3 heads of size 128, MLP units 32 with batch size 32. We train this architecture on each decoder

task individually and all tasks together for a total of 13 new models with the exact preprocessing steps as in the `P-Fusion` and `MultiModN` experiments. The results indicate that `MultiModN` often outperforms or at least matches the `P-Fusion` Transformer benchmark in the vast majority of single task and multi-task settings, and comes with several interpretability, missingness, and modularity advantages. Specifically, using the primary metric for each task (BAC for the classification tasks and MSE for the regression tasks), `MultiModN` beats the Transformer baseline significantly in 7 tasks, overlaps 95% CIs in 11 tasks, and loses very slightly (by 0.01) in 2 regression tasks.

**Classification**

| | Task | Encoder | BAC | F1 |
|---|---|---|---|---|
| MIMIC | Cardiomegaly | Single | 0.50 ± 0.01 | 0.43 ± 0.02 |
| | | Multi | 0.50 ± 0.02 | 0.43 ± 0.03 |
| | ECM | Single | 0.50 ± 0.02 | 0.44 ± 0.03 |
| | | Multi | 0.50 ± 0.01 | 0.42 ± 0.01 |
| EDU | Success (P/F) | Single | 0.50 ± 0.01 | 0.40 ± 0.01 |
| | | Multi | 0.96 ± 0.01 | 0.95 ± 0.02 |
| | Dropout | Single | 0.83 ± 0.22 | 0.83 ± 0.24 |
| | | Multi | 0.28 ± 0.02 | 0.27 ± 0.02 |

**Regression**

| | Task | Encoder | MSE | R2 |
|---|---|---|---|---|
| EDU | Weekly Perf. | Single | 0.01 ± 0.01 | 0.21 ± 0.06 |
| | | Multi | 3.91 ± 3.49 | -649.05 ± 506.01 |
| WEATHER | Temp. (24h) | Single | 0.00 ± 0.01 | 0.80 ± 0.10 |
| | | Multi | 0.00 ± 0.01 | 0.75 ± 0.08 |
| | Temp. (72h) | Single | 0.00 ± 0.01 | 0.71 ± 0.18 |
| | | Multi | 0.00 ± 0.01 | 0.77 ± 0.08 |
| | Temp. (720h) | Single | 0.00 ± 0.01 | 0.82 ± 0.13 |
| | | Multi | 0.00 ± 0.01 | 0.78 ± 0.03 |
| | Humidity | Single | 0.01 ± 0.01 | 0.69 ± 0.22 |
| | | Multi | 0.03 ± 0.01 | 0.00 ± 0.41 |
| | Visibility | Single | 0.02 ± 0.01 | 0.82 ± 0.11 |
| | | Multi | 0.05 ± 0.02 | 0.50 ± 0.12 |

Figure 14: **Performance of the `P-Fusion` Transformer on 10 classification and regression tasks across 3 datasets.** Results are showcased with 95% confidence intervals. BAC and MSE are the primary evaluation metrics for classification and regression respectively.

## E.5 Additional Inference Settings

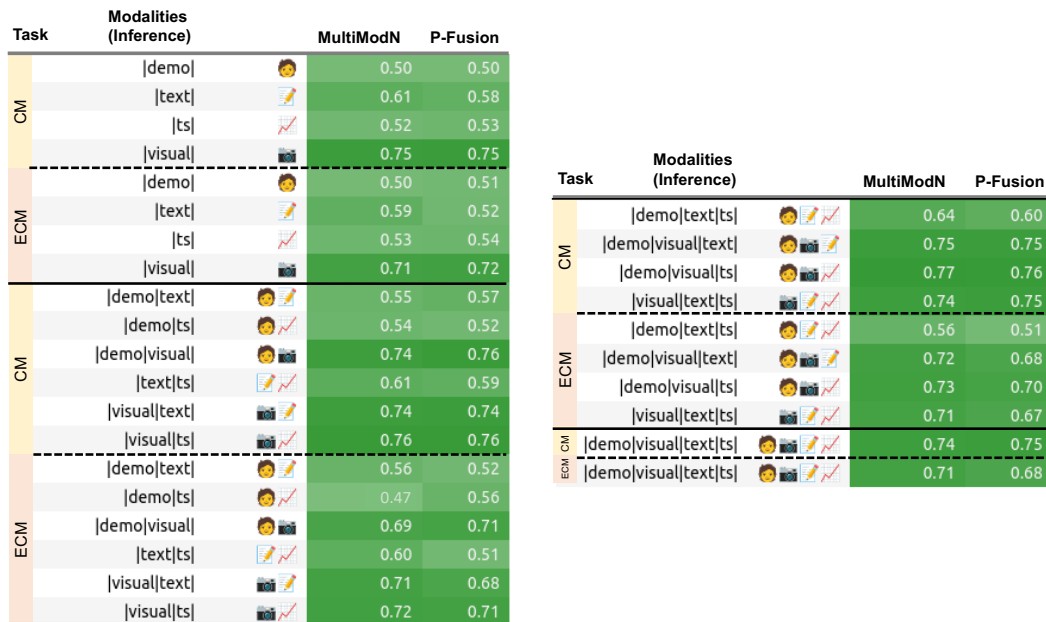

Figure 15: **Detailed modality inference experiments for `MultiModN` in comparison to `P-Fusion`.** In these experiments, different combinations of modalities and orderings at the time of inference are used for the two tasks in the MIMIC dataset. All 95% CIs overlap between the two models.

To provide insight into performance gains, we performed additional experiments to showcase the benefits of modularity with vastly different training and inference settings. The results of 30 new experiments of inference encoders, each performed with 5-fold cross-validation are included in Figure 15. We compare `P-Fusion` and `MultiModN` on both tasks of the MIMIC dataset using all possible combinations of four input modalities at test time. `MultiModN` ignores missing modalities whereas `P-Fusion` imputes and therefore encodes missing modalities.

We note that the performance at inference for `P-Fusion` and `MultiModN` has no significant differences for all experiments (using 95% CIs). Figure 15 shows that, on average, `P-Fusion` tends to overfit more to the most dominant (visual) modality. When this modality is missing (at random or completely at random), `MultiModN` performs better on a combination of the remaining modalities (demo, text, time series). In the case of missing modalities, the observed effect in Figure 15 is weak – confidence intervals overlap. Considering the MNAR (missing not-at-random) scenario described in Sec. 6.4, the difference becomes significant.

