# OpenReview forum: "MultiMoDN—Multimodal, Multi-Task, Interpretable Modular Networks"
_NeurIPS.cc/2023/Conference — NeurIPS 2023 poster_

### Official Review · Reviewer_mEsN · 2023-06-16

**Soundness:** 4 excellent
**Presentation:** 4 excellent
**Contribution:** 3 good
**Rating:** 7
**Confidence:** 4

**Summary:**

This paper presents MultiModN, a modular network that can deal with multimodal multitask problems and is inherently interpretable. MultiModN architecture consists of one encoder module for each modality, and one decoder module for each task. Each encoder module takes in a previous state and one modality input, and outputs the next state; the final state is obtained by sequentially passing the state through the encoder for each available modality (therefore this model can deal with data points with missing modalities as well).  The authors conducted experiments on 10 tasks across 3 real-world domains with vastly different modalities. MultiModN was compared to P-fusion, and both models used the exact same preprocessing steps and feature extraction to ensure fairness. The experiments showed that MultiModN is able to achieve similar performance compared to P-fusion while being inherently interpretable. Moreover, the authors conducted additional experiments that intentionally correlates missing modalities and certain labels in the training set, and shows that MultiModN is less affected by the correlation than P-Fusion (i.e. more robust against MNAR).

**Strengths:**

1. This paper studies the important problem of building robust and interpretable models for solving multimodal real-world tasks. Although the new model did not improve performance, improving model interpretability and robustness to spurious correlations is also extremely important for real-world settings where mistakes can have real impacts.

2. The experiments are done in a very well controlled setting. The authors purposefully used the exact same module architectures and feature extractors as the baseline to ensure fair comparison, and confidence intervals are always included. The experiments convincingly showed that the model is indeed inherently interpretable, and that the model is robust to MNAR compared to P-fusion and does not perform worse than P-fusion overall.

3. The paper presentation is generally quite good. The methodology description is clear and detailed, and the experiment details are all included. The figures and tables are nice looking and easy to understand.

**Weaknesses:**

The experiments are restricted to the exact same settings and pre-processing steps as P-fusion. While this ensures the fairness of comparison, it also means that we don't have evidence on whether this modular approach will also be able to achieve similar performance or alleviate MNAR problem if we have a different baseline approach / pre-processing. Perhaps a small proof-of-concept experiment showing that this approach can also work under different settings will greatly improve the significance and impact of this work.

**Questions:**

Q1.  Since all encoders are dense-layers (I assume that means MLPs), is it correct that the inputs to the encoders are the features extracted from each modality, which is always a single vector?

Q2. The only part of the paper that I am completely lost on is line 149-151 "Extension to time series". I understand that the M encoders are applied sequentially, but what does that have to do with time series? For time-series modality, doesn't the feature extractor turn it into a single vector such that it can go through the dense layers of the encoders (as in Q1)? I also don't see how this "extension" is relevant to any other part of the paper.

**Limitations:**

The authors have adequately addressed the limitations.

I believe there is no foreseeable potential negative social impact in this work.

---

> ### Author Rebuttal · Authors · 2023-08-09
>
> **SUMMARY:**
> We kindly thank the reviewer for this careful evaluation of our work and greatly appreciate the positive review. We strongly agree with all points raised.
> We have addressed each in a point-by-point response below and feel that the resulting edits have further improved the manuscript.
>
> **STRENGTHS:**
> We are happy that the reviewer feels the paper addresses the important problem of multimodality in real-world settings. We are additionally encouraged that the reviewer believes the experiments are conducted in a “well-controlled setting” and that the paper presentation is strong with “clear and detailed descriptions”. We are immensely grateful that reviewer 3 recognizes the contributions of the paper as we intended!
>
> **WEAKNESSES:**
> The reviewer rightly explains that our **intentional alignment of architectures (to ensure fair comparison) purposely limits our ability to make a performance comparison to the baseline**, and views this as a weakness.
>
> To provide insight into performance gains, we performed additional experiments as suggested to showcase the benefits of modularity with vastly different training and inference settings. The results of 30 new experiments of inference encoders, each performed with 5-fold cross-validation are included in Figure 1 of the rebuttal. We compare P-Fusion and our approach on both tasks of the MIMIC dataset using all possible combinations of four inputs modalities at test time. MultiModN ignores missing modalities whereas P-fusion imputes and therefore encodes missing modalities.
>
> We note that the performance at inference for P-Fusion and MultiModN has no significant differences for all experiments (using 95% CI). Figure 1 shows that, on average, P-Fusion tends to overfit more to the most dominant (visual) modality. When this modality is missing (at random or completely at random), Multi-ModN performs better on a combination of the remaining modalities (demo, text, time series). In the case of missing modalities, the observed effect in Figure 1 is weak – confidence intervals overlap. Considering the MNAR (missing not at random) scenario described in Experiment 3 of the paper (Section 6.3), the difference becomes significant.
>
> **QUESTIONS:**
> The reviewer had 2 questions:
>
> **Regarding the inputs of encoders**: yes in our case, the input is always a single vector and the encoders are always MLP. For the MIMIC dataset that contains images as an input modality, we use the pre-trained CNN to preprocess the data, which results in a vector. But nothing in the design of MultiModN restricts us to the use of MLP encoders that expects a 1D vector as an input. The pre-trained CNN could be included in the encoder as a frozen part, so that the encoder would receive images as input without any changes in the operations applied on the input. Any suitable architecture can be used as an encoder. We chose the presented simplified architecture to demonstrate the approach. We agree that this point needs to be clarified and made explicit. We will state explicitly that in our experiments, all encoders are MLP that expect 1D vectors as input.
>
> The second question of the reviewer **discusses the role and relevance of the "extension to time series” section** in the architecture. We agree that the description of the time series setting of the MultiModN architecture (relevant in EDU and Weather) can be better described.  As the reviewer mentioned correctly, in the MIMIC setting, we show adding the time-series directly as a modality, which embeds the data as a single vector.
>
> However, in real-world settings with data streams, real-time data is collected continuously over weeks and months at the same time the model is expected to inference. For example, the education version of MultiModN can make a prediction of student pass-fail success at week 1, week 2, week 3, etc. with new data from each week. MultiModN allows us to simply add the new week’s data (i.e. week 3) and update the predictions on all tasks (without requiring the model to inference again on week 1 or week 2 predictions). As described in the appendix, the “continuous” classification or regression tasks (Tasks 4 through 8) predict at every timestep with the new data from that timestep (each hour for the temperature forecasting and each week for the student success or dropout prediction).
>
> **LIMITATIONS:**
> We are happy that the reviewer feels we have appropriately addressed all limitations in the paper.

---

> > ### Comment · Reviewer_mEsN · 2023-08-15
> >
> > Thank you for your clarifications and the additional studies regarding applying your method to different settings! I raised my score from 6 to 7.

---

### Official Review · Reviewer_iERT · 2023-07-05

**Soundness:** 2 fair
**Presentation:** 2 fair
**Contribution:** 2 fair
**Rating:** 4
**Confidence:** 3

**Summary:**

This paper describes an architecture for multimodal multitask learning which is robust to missing modalities/tasks both at training and test time. This is composed of:
- encoder specific modules
- (assuming an ordering among modalities) a hidden states which depends on the output of a modality specific encoder, and which feds into the next encoder,
- a decoder that takes the state and produced an output for each task.
The method is tested on three multimodal datasets, namely MIMIC, EDU and Weather2K.

**Strengths:**

- multimodal multitask learning is a very interesting topic which is very relevant to this venue
- the idea to make a multimodal multitask system robust to missing modalities is clearly good
- the overall motivation and intuition behind the proposed architecture is sensible and intuitive

**Weaknesses:**

- The major limitation is the lack of reference and comparison to other multimodal multitask approaches, particularly those based on transformers in the vision/language domain. For instance,
[1] UniT: Multimodal Multitask Learning with a Unified Transformer
Ronghang Hu Amanpreet Singh CVPR 2021
[2] Are Multimodal Transformers Robust to Missing Modality?
Mengmeng Ma, Jian Ren, Long Zhao, Davide Testuggine, Xi Peng CVPR 2022
The submission would be much stronger if the authors would discuss their contribution relative to these works, and possibly compare empirically their architecture to these ones using the same benchmarks
- Related to the point above, there should be at least discussion about the recent trend to turn every task to text generation using a large language model (LLM), and multimodal multitask system leveraging LLMs. For instance:
[3] Flamingo: a Visual Language Model for Few-Shot Learning
Jean-Baptiste Alayrac, Jeff Donahue et al. NeurIPS 2022
- Related to transformers, the choice of using a RNN seems rather unnatural because it enforces an artificial ordering over modalities. Instead transformers operate on sets (of any size). I wonder whether the authors have considered replacing the RNN with a transformer.
- The paper lacks clarity. The method should be described in a more formal way to better understand the implementation details. For instance, I am unclear:
  - whether decoder parameters are shared across the different modalities,
  - how the decoders predictions are combined across modalities/modules,
  - what parameters are actually subject to training,
  - what happens in the encoder when a modality is missing
- The empirical results are weak. In fact, the proposed method often times works worse than the simple fusion baseline.
- It would be nice if the authors demonstrated the benefits of modularity, for instance by:
  - adding a new task or modality over time
  - improving a task module with a subset of modalities at training time, the task improves also when at test time other modalities are used, etc.

**Questions:**

See comments above in the weakness section.
The major question is about 1) relation to prior work, particularly in the vision/language domain, 2) use of more standard benchmarks (in addition to the chosen ones), 3) use of RNN as opposed to transformer, 4) various clarifications of the approach.

**Limitations:**

No concern.

---

> ### Author Rebuttal · Authors · 2023-08-09
>
> **SUMMARY:**
> We kindly thank the reviewer for the careful evaluation of our work and greatly appreciate the review. We address each concern in the point-by-point response below and feel that the resulting edits have further improved the manuscript.
>
> **STRENGTHS:**
> We are happy that the reviewer feels the paper covers an interesting topic that is very relevant to NeurIPS. We are also glad that the reviewer agrees on the importance of making multimodal models resilient to the common issue of systematic missingness and that they find our approach sensible and intuitive.
>
> **WEAKNESSES:**
> We agree with the reviewer’s opinion that our submission would be much stronger if compared to **further multimodal multitask baselines, particularly transformers in the vision/language domain**. We had actually tested several baselines (e.g., BiLSTMs, various compositions of P-Fusion), none of which improved upon MultiMoDN. For readability, they were excluded. We now include a transformer baseline selected with hyperparameter tuning on all 3 datasets. The results are found in Table 1 of the PDF and will be included in the final submission.
>
> The results indicate that MultiModN outperforms or at least matches the transformer benchmark in the vast majority of single and multitask settings, and comes with several interpretability, missingness, and modularity advantages. Specifically, using the primary metric for each task (BAC for classification and MSE for regression tasks), MultiModN beats the transformer baseline significantly in 7 tasks, overlaps 95% CIs in 11 tasks, and loses very slightly (by 0.01) in 2 regression tasks.
>
> We agree with the reviewer that a **larger discussion on the recent multimodal trends using transformers and LLMs** would improve the paper. UNiT is a promising multimodal, multitask transformer architecture, however, it remains monolithic, trained on the union of all inputs (padded when missing) inputted into the model in parallel. This risks exposing the model to systematic missingness during training and reduces model interpretability (requiring all modalities to be represented even if not present) and portability (the transformer has 427,421 trainable params for EDU while MultiModN achieves better performance with 12,159). [1]’s recent work has found similar results on the erratic behavior of transformers to missing modalities, but is only tested on visual/text inputs.
>
> The recommendation of the LLM approach Flamingo is interesting but is also limited to only 2 modalities (visual/text). It is not clear how tabular and time series would be handled or how this would affect the context window at inference. Combining predictive tasks with LLMs will also greatly impact interpretability, introducing hallucinations and creating a model complexity that may use learned text bias to influence predictions.
>
> **The reviewer wonders whether transformers are superior as they don’t impose an order which may be “unnatural”.** MultiMoDN can be completely order-invariant and idempotent as shown in [2] and only requires randomization during training to achieve this. For interpretability, we argue that sequential inference (in any order) is far superior to parallel input due to its decomposability: allowing the user to visualize the effect of each input and aligning with Bayesian reasoning.
>
> We will present this improved discussion of our positioning in relation to recent multimodal transformers and LLMs in the final paper.
>
> The reviewer makes excellent recommendations to improve the **clarity of our paper and the formalism of our architecture**. We have integrated all of them, specifically detailing that:
>   - Decoder parameters are indeed shared across the different modalities.
>   - The decoder predictions are combined across modalities/modules by averaging the loss. It is interesting to note that a weighted loss scheme could force the model to emphasize certain tasks over others.
>   - All encoder and decoder parameters are subject to training, except for the pretrained encoders used to generate embeddings in the MIMIC dataset (used to replicate the exact baseline parallel fusion setting).
>   - When a modality is missing, the encoder is skipped and not trained. (This is also depicted in Figure 1 of the original submission).
>
> The reviewer claims that the **empirical results are weak because our model is not superior to the baseline in terms of performance**.
> It is critical to understand that we (purposely) do not aim to improve on the baseline in terms of performance. Rather, we limit the model’s ability to do so, by aligning the pre-extracted features to isolate the comparison of the fusion step. Multimodal multi-task models often have degraded performance compared to their single modality counterparts [3]. We argue that our model is by far superior to the baseline by virtue of being modular, interpretable, composable, robust to systematic missingness and multi-task without impacting performance.
>
> The reviewer would like to see more support for our method’s performance using **various numbers and combinations of inputs at inference**. This is an excellent suggestion. We now include the results of our model on various numbers and combinations of inputs (see Figure 1 of the attached PDF). The baseline would have to impute missing features in all of these combinations, exposing it to catastrophic failure in the event of systematic missingness (Section 6.3).
>
> **LIMITATIONS:**
> We are happy that the reviewer feels we have appropriately addressed all limitations in the paper.
>
> **REFERENCES:**
> [1] Ma, Mengmeng, et al. "Are multimodal transformers robust to missing modality?" CVPR 2022.
> [2] Trottet, Cecile, et al. "Modular Clinical Decision Support Networks (MoDN)—Updatable, interpretable, and portable predictions for evolving clinical environments." PLOS 2023.
> [3] Liu, Shengchao, et al. "Loss-balanced task weighting to reduce negative transfer in multi-task learning." AAAI 2019.

---

> > ### Comment · Reviewer_iERT · 2023-08-16
> > **post-rebuttal assessment**
> >
> > I would like to thank the authors for their response, and additional material.
> > The authors have clarified several questions and provided stronger empirical support for their approach, and I have raised my rating for this submission. While I am not opposed to accepting this work, I still feel it is borderline because a major revision would be needed to address and integrate in the submission all the points of this discussion. Even though the empirical validation does not include popular language-X domains, this is however now sufficient to support the proposed approach.

---

> > > ### Author Response · Authors · 2023-08-21
> > >
> > > We thank R2 for their response. We are happy that stronger experimental support for MultiModN has been conveyed and that the questions raised by the reviewer have been fully answered. To the best of our knowledge, **we have addressed all of the reviewer’s concerns**.
> > >
> > > Note that all experiments and analyses have **already been completed** and all discussion intended to be included in the paper has **already been written** into the rebuttal. We therefore expect this will only be a **minor revision**.
> > >
> > > For your perusal, we have already included in the following comment **the changes we intend to make within the allowed additional page**.

---

> > > > ### Author Response · Authors · 2023-08-21
> > > >
> > > > **Emphasizing novelty of work over baselines**
> > > > **[End of the Introduction] [Line 80]**
> > > >
> > > > Note that our experimental setup purposely limits our model performance to fairly compare the multimodal fusion step. At equivalent performance, our model architecture is by far superior to the baseline by virtue of being inherently modular, interpretable, composable, robust to systematic missingness, and multi-task.
> > > >
> > > > **Discussing the recent related Multimodal work with Transformers, LLMs**
> > > > **[End of the Background section on Parallel Fusion (2.1)] [Line 108]**
> > > >
> > > > Several other recent architectures utilize parallel fusion with transformers. UNiT (Unified Transformer) [1] is a promising multimodal, multitask transformer architecture; however, it remains monolithic, trained on the union of all inputs (padded when missing) inputted in parallel. This not only exposes the model to patterns of systematic missingness during training but also reduces model interpretability and portability\footnote[The equivalent transformer architecture has 427,421 trainable params for EDU while MultiModN achieves better performance with 12,159]. [2]’s recent work has found similar results on the erratic behavior of transformers to missing modalities, although it is only tested on visual/text inputs. LLMs have also recently been used to encode visual and text modalities [3], but it is not clear how tabular and time series would be handled or how this would affect the context window at inference. Combining predictive tasks with LLMs will also greatly impact interpretability, introducing hallucinations and complex predictive contamination where learned textual bias can influence outcomes.
> > > >
> > > >
> > > > **Clarifying the architecture details**
> > > > **[Throughout Architecture Section 4]**
> > > >
> > > > **[Line 151]** This is relevant in the real-world setting of a data stream, where inference takes place at the same time as data is being received (i.e. predicting student performance at each week of a course as the course is being conducted). The continuous prediction tasks (shown for EDU and Weather in Section 6) demonstrate how MultiModN can be used for incremental time series prediction.
> > > >
> > > > **[Line 157]** All encoder and decoder parameters are subject to training, except for the pretrained encoders used to generate embeddings in the MIMIC dataset (used to replicate the exact baseline parallel fusion setting).
> > > >
> > > > **[Line 177]** The input vectors in our experiments are 1D. When a modality is missing, the encoder is skipped and not trained (depicted in Figure 1).
> > > >
> > > > **[Line 180]** Decoder parameters are shared across the different modalities. The decoder predictions are combined across modalities/modules by averaging the loss. Interestingly, a weighted loss scheme could force the model to emphasize certain tasks over others.
> > > >
> > > > **[Line 182]** As shown in [4], MultiMoDN can be completely order-invariant and idempotent if randomized during training. For interpretability, sequential inference (in any order) is superior to parallel input due to its decomposability: allowing the user to visualize the effect of each input and aligning with Bayesian reasoning.
> > > >
> > > >
> > > > **Discussing the new transformer baseline results**
> > > > **[End of Experimental Section 6.1] [Line 281]**
> > > >
> > > > We provide an additional parallel fusion transformer baseline with experimental results showcased in Appendix E.4. The results indicate that MultiModN matches or outperforms the multimodal transformer in the vast majority of single and multitask settings, and comes with several interpretability, missingness, and modularity advantages. Specifically, using the primary metric for each task (BAC for classification and MSE for regression tasks), MultiModN beats the transformer baseline significantly in 7 tasks, overlaps 95% CIs in 11 tasks, and loses very slightly (by 0.01) in 2 regression tasks.
> > > >
> > > > **Referencing additional inference experiments in appendix**
> > > > **[End of Experimental Section 6.2] [Line 297]**
> > > >
> > > > We additionally include the results of our model on various numbers and combinations of inputs, described further in Appendix E.5. The baseline would have to impute missing features in these combinations, exposing it to catastrophic failure in the event of systematic missingness (Section 6.3).
> > > >
> > > >
> > > > [1] Hu, Ronghang, and Amanpreet Singh. "UniT: Multimodal multitask learning with a unified transformer." Proceedings of the IEEE/CVF International Conference on Computer Vision. 2021.
> > > > [2] Ma, Mengmeng, et al. "Are multimodal transformers robust to missing modality?" CVPR 2022.
> > > > [3] Alayrac, Jean-Baptiste, et al. "Flamingo: a visual language model for few-shot learning." Advances in Neural Information Processing Systems 35 (2022).
> > > > [4] Trottet, Cecile, et al. "Modular Clinical Decision Support Networks (MoDN)—Updatable, interpretable, and portable predictions for evolving clinical environments." PLOS 2023.

---

### Official Review · Reviewer_pFnK · 2023-07-07

**Soundness:** 3 good
**Presentation:** 4 excellent
**Contribution:** 3 good
**Rating:** 7
**Confidence:** 3

**Summary:**

This paper proposes a modular multimodal model that fuses latent representations in a sequence of modality to conduct a combination of predictive tasks. It utilizes a flexible sequence of model and task-agnostic encoders to produce an evolving latent representation for a combination of multi-task, model-agnostic decoder modules to conduct downstream tasks. The authors conduct experiments on benchmarks of different domains to show the effectiveness of the proposed method. The experiments show that the proposed method outperforms traditional monolithic multi-modal models.

**Strengths:**

1. This paper is well-written and well-organized. The design of methods and experiments are clear and easy to understand.
2. The proposed method is novel and effective. The modules are task-agnostic and can be adapted on any number of tasks on different modalities.
3. The experiments are substantial and convincing. The proposed method shows significantly better performance compared with traditional methods.

**Weaknesses:**

1. It seems that for single tasks the proposed method cannot provide improvement compared with the baseline.
2. It would be helpful to support the claim that the proposed model can handle  any number/combination of tasks by adding more experiments of combinatorial tasks.

**Questions:**

1. The authors should make the figures in the experiments part more clear to see the comparison of the proposed methods and the baseline.

**Limitations:**

The authors have addressed the technical limitations in the paper. I do not find any negative societal impact of their work.

---

> ### Author Rebuttal · Authors · 2023-08-09
>
> **SUMMARY:**
> We kindly thank the reviewer for this careful evaluation of our work and greatly appreciate the positive review of our “excellent” presentation, methodological soundness, and worthwhile contribution.
>
> We strongly agree with all points raised. We have addressed each in a point-by-point response below and feel that the resulting edits have further improved the manuscript.
>
> **STRENGTHS:**
> We are happy that the reviewer feels the paper is well-written, well-organized, clear, and easy to understand. We are also thrilled to see that they appreciate the “novelty and effectiveness” of our proposed model, and that our experiments are substantial and convincing.
>
> **WEAKNESSES:**
> The reviewer notes that for single tasks, **our method does not improve compared with the baseline**.
> Indeed, this is intended. It is critical to understand that we (purposely) do not aim to improve on the baseline in terms of performance. Rather, we purposely limit the model’s ability to do so, by aligning the pre-extracted features. This is to isolate the comparison of the fusion step. In a phenomenon called “negative transfer”, multimodal multi-task models often have degraded performance compared to their single-modality counterparts [1].
>
> We argue that our model is by far superior to the baseline by virtue of being modular, interpretable, composable, robust to systematic missingness, and multi-task *without* impacting performance compared to the parallel baseline which does not have these other features and will make this point more explicit in the final manuscript.
>
> The reviewer would like to see **more support for our method’s performance using various numbers and combinations of inputs** at inference. This is an excellent suggestion. We now include additional experiments detailing the results of our model on various numbers and combinations of inputs (see Figure 1 of supplemental PDF information). The baseline would have to impute missing features in all these combinations, exposing it to catastrophic failure in the event of systematic missingness as explained in Section 6.4 and supplemental Section E.3.
>
> **QUESTIONS:**
> Thank you for the suggestion to improve the experimental figures for clearer understandability. As recommended, we have made several formatting edits to better distinguish MultiMoDN and the baseline directly in the figures. The revised figures are showcased in the attached PDF, with differences highlighted in yellow.
>
> **LIMITATIONS:**
> We are happy that the reviewer feels we have appropriately addressed all limitations in the paper.
>
> **REFERENCES:**
> [1] Liu, Shengchao, Yingyu Liang, and Anthony Gitter. "Loss-balanced task weighting to reduce negative transfer in multi-task learning." AAAI 2019.

---

> > ### Comment · Area_Chair_M2Vj · 2023-08-19
> >
> > Hi Reviewer pFnK,
> >
> > Since the discussion with the authors is closing soon, could you please go over the author's rebuttal and provide some feedback?
> >
> > Regards,
> > AC

---

### Author Rebuttal · Authors · 2023-08-09

We are grateful to the reviewers for their careful evaluation and very happy to receive positive and high-quality reviews. We agree with all points raised and have been able to reflect and respond to each in detail.

We highlight a major concern of the only reviewer (R2) recommending rejection: our model does not show significant performance improvement compared with the baseline. This is an important misunderstanding, as we actually purposely limit our model performance to fairly compare the multimodal fusion step. At equivalent performance, our model architecture is by far superior to the baseline by virtue of being inherently modular, interpretable, composable, robust to systematic missingness, and multi-task. The aim was to achieve these advantages without impacting performance compared to the baseline, which does exhibit these advantages.

Indeed, R3 cites this approach as an important strength: *“The experiments are done in a very well controlled setting. The authors purposefully used the exact same module architectures and feature extractors as the baseline to ensure fair comparison, and confidence intervals are always included. The experiments convincingly showed that the model is indeed inherently interpretable, and that the model is robust to MNAR compared to P-fusion and does not perform worse than P-fusion overall.”*

**R1** provides a very positive review and has recommended acceptance, citing our *“convincing and sound experiments”* described in a *“well-written, organized, and clear”* manner. They have requested the following:
1. **More support for our method’s ability to use various numbers and combinations of inputs at inference.** We provide 30 new inference experiments in Figure 1 in the attached PDF. The experiments test all combinations of different inference settings from the training setting and show the expected/desired outcome, where our method is not significantly different from the baseline (all 95% CIs overlap) despite having the various advantages of modularity (robustness to systematic missingness, interpretability, portability).
2. **Further clarity on the experimental figures.** We have formatted Figures 3, 4, and 5 and Appendix Figure 11 to better highlight the difference between our model and the baseline. These improved figures are included as Figure 2 in the attached PDF (changes highlighted in yellow).

**R2** provides a generally positive review (*“interesting topic”*, *“relevant to NeurIPS”*, *“sensible and intuitive approach”*),  but recommends rejection. They cite several issues motivating this recommendation which have now been fully addressed:
1. **The reviewer recommends a transformer baseline and a commentary on multi-modal LLM approaches.** Based on the related work mentioned by R2, we discuss in detail how our work compares to other approaches (in vision and text). Notably, we have conducted additional experiments to now include a multitask, multimodal transformer baseline (found in Table 1 of the attached results), which is overall less performant than our architecture and does not have the advantages of interpretability or modularity. We further explain how LLMs are not well suited to the task at hand, as they greatly degrade interpretability and introduce the risk of hallucination from previously biased text inputs.
2. **The reviewer asks for various clarifications regarding the model architecture** regarding decoder parameter sharing, decoder loss, trained parameters, and treatment of missing modalities. We have elaborated on each of these areas formally in our architecture section and would like to thank the reviewer for their careful perusal.

**R3** provided a positive review stating *“improving model interpretability and robustness to spurious correlations is extremely important for real-world settings where mistakes can have real impacts”* and recommends acceptance:
1. **The reviewer requests that we experiment with diverse train and test settings** beyond an exact comparison with P-Fusion, aligning with R1. Our new experiments in Figure 1 address this concern.
2. **The reviewer requests architecture clarifications on the inputs of the encoders and the “extension to time series” section**. The input vectors in our experiments are 1D. Depending on the nature of the modalities in the dataset, we support other encoders (CNN, transformers, and other pretrained models) to determine the ideal encoding for each input/modality. For time-series, we present the real-world setting of a data stream, where inference takes place at the same time as data is being received (i.e. predicting student performance at each week of a course as the course is being conducted). These settings are relevant for continuous prediction tasks in EDU and Weather, showing how MultiModN can be used for incremental time series prediction. We will state these descriptions explicitly in the paper.

To summarize the additional experiments, we provide the following results to strengthen our paper in the PDF attachment:
- **New results on all 10 tasks across 3 datasets from a multimodal multitask transformer architecture** based on the related work highlighted by R2 (Table 1). These results show that MultiModN is comparable or better than the Transformer in both single task and multi-task settings, in addition to having advantages that are not present with the Transformer (size, modularity, interpretability).
- **30 new inference modality experiments comparing P-Fusion and MultiModN in different settings and tasks** (Figure 1). These indicate comparable or often better performance in contrast with P-Fusion across diverse inference settings.
- **Improved diagrams from the results sections** to clearly understand the difference between the baseline and MultiModN (Figure 2).

Overall, we thank the reviewers for their thoughtful and expert advice. We hope that we have adequately addressed reviewer concerns and further improved confidence in our work.

---

> ### Author Response · Authors · 2023-08-19
>
> Dear reviewers and AC,
>
> We are glad that our additional contributions have addressed your concerns. Thank you for seeing the value in our work and increasing your scores towards acceptance. We hope that the positive scores and especially the positive opinions of our reviewers have convinced the AC.
>
> Thanks,
> MultiModN Authors

---

### Decision · Program_Chairs · 2023-09-21

**Decision:**

Accept (poster)

**Comment:**

The paper proposes a new multimodality model that leverages a sequential design to process different modalities. Reviewers generally appreciate the thoroughly controlled experiments, the importance of the problem, and the intuitive architecture design. The concern of reviewer iERT also seems to have been addressed in the rebuttal.